# Adult mouse fibroblasts retain organ-specific transcriptomic identity

Elvira Forte[1]*[†], Mirana Ramialison[2,3,4†], Hieu T Nim[2,3,4†], Madison Mara[1], Jacky Y Li[4], Rachel Cohn[5], Sandra L Daigle[1], Sarah Boyd[6], Edouard G Stanley[4,7,8], Andrew G Elefanty[4], John Travis Hinson[5,9], Mauro W Costa[1,2], Nadia A Rosenthal[1,2,10‡], Milena B Furtado[1,2]*[‡§]

[1]The Jackson Laboratory, Bar Harbor, United States; [2]Australian Regenerative Medicine Institute, Monash University, Clayton, Australia; [3]Systems Biology Institute Australia, Clayton, Australia; [4]Murdoch Children's Research Institute, The Royal Children's Hospital, Parkville, Australia; [5]The Jackson Laboratory, Farmington, United States; [6]Centre for Inflammatory Diseases, School of Clinical Sciences at Monash Health, Monash University, Clayton, Australia; [7]Department of Paediatrics, Faculty of Medicine, Dentistry and Health Sciences, University of Melbourne, Parkville, Australia; [8]Department of Anatomy and Developmental Biology, Monash University, Clayton, Australia; [9]Cardiology Center, UConn Health, Farmington, United States; [10]National Heart and Lung Institute, Imperial College London, London, United Kingdom

**\*For correspondence:**
elviraforte83@gmail.com (EF);
milenabfurtado@gmail.com
(MBF)

[†]These authors contributed
equally to this work
[‡]These authors also contributed
equally to this work

**Present address:** [§]Merck
Research Laboratories, South San
Francisco, United States

**Reviewing Editor:** Paul W
Noble, Cedars-Sinai Medical
Center, United States

**Abstract** Organ fibroblasts are essential components of homeostatic and diseased tissues. They participate in sculpting the extracellular matrix, sensing the microenvironment, and communicating with other resident cells. Recent studies have revealed transcriptomic heterogeneity among fibroblasts within and between organs. To dissect the basis of interorgan heterogeneity, we compare the gene expression of murine fibroblasts from different tissues (tail, skin, lung, liver, heart, kidney, and gonads) and show that they display distinct positional and organ-specific transcriptome signatures that reflect their embryonic origins. We demonstrate that expression of genes typically attributed to the surrounding parenchyma by fibroblasts is established in embryonic development and largely maintained in culture, bioengineered tissues and ectopic transplants. Targeted knockdown of key organ-specific transcription factors affects fibroblast functions, in particular genes involved in the modulation of fibrosis and inflammation. In conclusion, our data reveal that adult fibroblasts maintain an embryonic gene expression signature inherited from their organ of origin, thereby increasing our understanding of adult fibroblast heterogeneity. The knowledge of this tissue-specific gene signature may assist in targeting fibrotic diseases in a more precise, organ-specific manner.

## Editor's evaluation

The authors aim to show that fibroblasts have a heterogenous transcriptome that is retained throughout their lifetime due to their source of embryonic origin. Of great interest is that compelling evidence is provided that these transcriptional signatures have direct translational consequences. This is shown through coculture experiments, where coculture of cardiomyocytes with non-cardiac fibroblasts impairs integration and contractility, while cardiac fibroblasts integrate with cardiomyocyte cultures to create functional beating tissue. This memory is shown to be malleable: three days post implantation in the renal capsule, explanted fibroblasts largely maintained their original transcriptomic signature, while also showing the onset of adaptation to a new microenvironment. In

addition, markers are identified which allow the separation of fibroblasts based on their anatomical origin. Considering the lack of tissue-specific markers for fibroblasts, this is a significant advance.

## Introduction

Fibroproliferative disorders are the main cause of mortality and morbidity in developed countries, accounting for about 45% of deaths in the United States (*Bitterman and Henke, 1991*). Despite the impactful prevalence of chronic organ fibrosis, current antifibrotic drugs are both inefficient and nonspecific to this condition (*Friedman et al., 2013*; *Henderson et al., 2020*). Fibroblasts, main players in fibrosis, have gained increased attention for their capacity to perform functions far beyond their canonical secretion of extracellular biological scaffolding and formation of scar tissue after injury. Recent literature poses the organ fibroblast as a major regulatory hub that senses local microenvironment imbalances and controls tissue remodeling (*Gerber et al., 2018*) upon activation and phenotypic differentiation into the profibrotic myofibroblast (*Pakshir et al., 2020*). They are also involved in immunomodulation (*Van Linthout et al., 2014*), by producing and responding to cytokines that activate immune cells of the innate and adaptive immune systems (*Forte et al., 2018*; *Boyd et al., 2020*), through organ-specific regulatory networks (*Krausgruber et al., 2020*).

Organ fibroblasts have been historically difficult to identify and study in vivo, due to their vague functional definition and lack of adequate markers that label organ fibroblast pools completely and specifically (*Swonger et al., 2016*). Recent advances in lineage tracing and multiomics single-cell analyses have revealed a significant heterogeneity of fibroblasts within and among tissues, and we are just beginning to understand how fibroblast heterogeneity correlates with distinct functions (*Henderson et al., 2020*; *Shaw and Rognoni, 2020*; *Lynch and Watt, 2018*; *LeBleu and Neilson, 2020*; *Griffin et al., 2020*). Despite being morphologically similar, spindle-shaped mesenchymal cells located in stromal tissues, fibroblasts acquire specialized functions related to their anatomical position (*Krausgruber et al., 2020*; *Slany et al., 2014*; *Foote et al., 2019*). They appear to retain a positional memory of the embryonic developmental axis: anterior–posterior, proximal–distal, and dermal–nondermal, possibly reflecting their role in conveying positional identity in embryogenesis (*Rinn et al., 2006*; *Rinn et al., 2008*; *Ackema and Charité, 2008*; *Chang et al., 2002*). The retention of positional molecular information through to adulthood suggests organ fibroblasts respond to molecular cues that drive body compartmentalization. Fibroblast heterogeneity within an organ tends to arise from the distinct embryological origin and/or anatomical localization (*Lynch and Watt, 2018*; *Griffin et al., 2020*; *Muhl et al., 2020*; *Forte et al., 2020*), while interorgan differences have been mostly ascribed to the matrisome, as shown by the transcriptomic comparison among fibroblasts from muscular tissues (*Muhl et al., 2020*).

Having previously reported that fibroblasts isolated from the adult mouse heart retain a cardiogenic transcriptional program (*Furtado et al., 2014a*), we show here that fibroblasts isolated from different adult organs similarly retain the expression of transcription factors and other gene sets involved in the determination of organ formation and patterning during embryonic development. This signature is captured in nascent embryonic organ fibroblasts and retained in adult fibroblasts under culture in isolation or in coculture with parenchymal cells from different organs. Further ectopic transplantation of fibroblasts into a different organ in vivo demonstrates the strength of the organ molecular signature despite new microenvironmental challenges. The robustness of the fibroblast organ transcriptome signature shown here supports its importance for organ interaction, connectivity, and function. In addition, knockdown of selected organ development transcription factors in cardiac fibroblasts deregulated the expression of genes involved in inflammation, fibrosis, and extracellular matrix (ECM) deposition, further supporting the relevance of these genes in fibroblasts function. In summary, our study uncovers stable expression of organ-specific, development-related signature genes in adult fibroblasts, thus offering new prospects for possible antifibrotic therapies.

## Results

### Cell homeostasis and ECM components comprise a generic fibroblast gene signature

To compare the gene signature of fibroblasts from different organs and eliminate potential RNA contaminants from other organ cell types, dissociated adult murine tissues were cultured for 5 days, followed by sorting for CD45−CD31−CD90+ fibroblasts (*Furtado et al., 2014a*; *Figure 1—figure supplement 1a*). High-throughput gene expression profiling identified 1281 highly expressed genes common to all fibroblast types, comprising the generic fibroblast signature (*Figure 1—source data 1*).

Through Ingenuity Pathway Analysis (IPA, Qiagen), we classified the commonly expressed genes based on cellular function (*Figure 1—figure supplement 1b*) and cellular localization (*Figure 1—figure supplement 1c*). Top functions included mechanisms of cell maintenance, such as proliferation, cytoskeletal arrangement, and cell movement, as well as general metabolic processes, including carbohydrate, nucleic acid protein, and small molecule biochemistry. Common fibroblast identifier genes were encountered within various IPA process classification groups. As an example, the cell surface receptor CD90 (*Thy1* gene) belongs with cellular assembly and organization, growth and proliferation and protein synthesis, while the myofibroblast marker smooth muscle actin (*Acta2* gene) was found in functions of cellular movement. The presence of CD90 and absence of CD31 (*Pecam1* gene)/CD45 (*Ptprc* gene) in all organ groups validated our positive/negative selection strategy for cell isolation, indicating a generally consistent population of cells in all organs. ECM elements, including

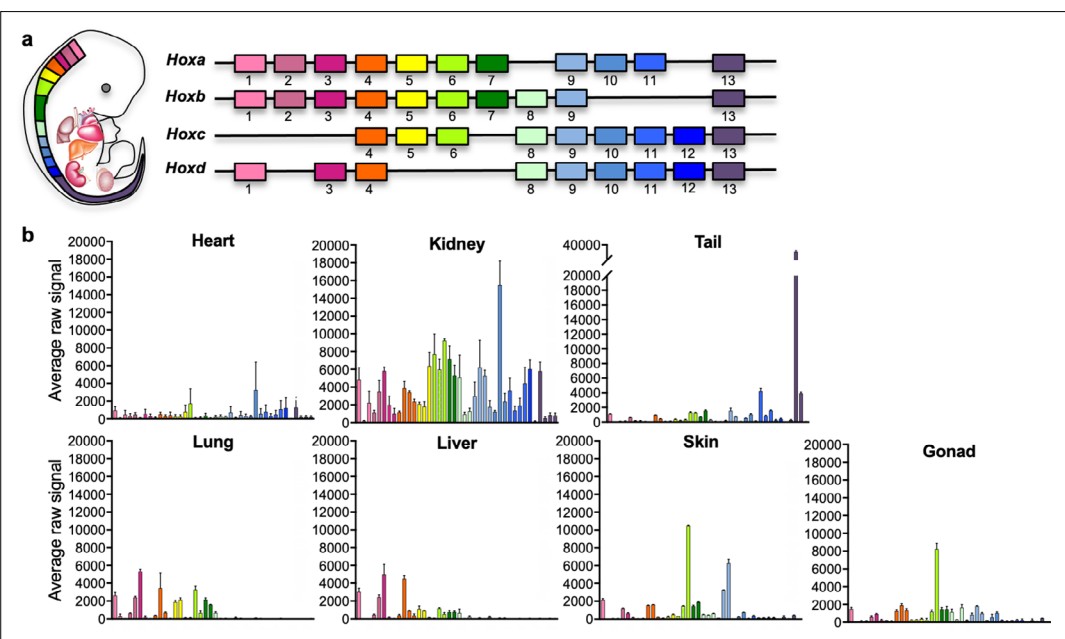

**Figure 1.** Positional code of organ fibroblasts. (**a**) Murine *Hox* cluster code, showing proximally expressed *Hox* genes in pink and distal ones in purple. (**b**) Bar plots showing the average raw signal for all expressed *Hox* genes in organ-specific fibroblast samples. Data are mean ± standard error of the mean (SEM) of three biological replicates for each fibroblast type from the microarray analysis. Refer to *Figure 1—figure supplements 1–2*, *Figure 1—source data 1*, and *Figure 1—source data 2*.

The online version of this article includes the following source data and figure supplement(s) for figure 1:

**Source data 1.** Microarray data: highly expressed genes common to all organ fibroblast populations, classified based on cellular process or cellular localization.

**Source data 2.** Microarray data: average raw expression and standard errors of *Hox* code genes across all fibroblast samples from the microarray analysis (*n* = 3).

**Figure supplement 1.** Isolation of fibroblasts from different organs and IPA analysis of common genes expressed across all organ fibroblasts.

**Figure supplement 2.** Pairwise comparison of *Hox* code across organ fibroblasts.

collagens, were included in several functional annotations, such as to cell morphology, assembly and organization, cellular compromise, function, and maintenance or cell signaling.

## Organ fibroblasts retain *Hox* codes

The *Hox* code defines body segmental identity and is highly conserved from flies to mammals. *Hox* genes show colinear expression and undergo chronological activation in the embryo, where upstream genes successively activate downstream genes in an anteroposterior fashion, such that upstream genes are activated first in more anterior segments of the body. In mammals, the *Hox* cluster has undergone a series of duplications and deletions that led to the formation of four paralogous clusters a, b, c, and d (*Figure 1a*).

Site-specific *HOX* expression has been previously reported in human skin fibroblasts (*Rinn et al., 2006*; *Rinn et al., 2008*; *Chang et al., 2002*) and mouse mesenchymal cells isolated from different organs (*Ackema and Charité, 2008*), and it has been shown to be cell autonomous and epigenetically maintained, suggesting a source of positional memory to differentially pattern tissue-specific homeostasis and regeneration. To determine if fibroblasts isolated from other adult mouse organs retain a distinct *Hox* signature, we plotted the average raw expression of all *Hox* genes per each fibroblast type (*Figure 1b*, *Figure 1—figure supplement 2*, *Figure 1—source data 2*). Among profiled organs, five patterns of *Hox* expression were identified: lung and liver showed expression of anterior *Hox* genes, in particular genes from the clusters 1–7, although liver had lower *Hox4–7* expression when compared with lung. A second group including skin and gonad displayed high *Hox6* expression, with skin from thoracic and abdominal ventral skin areas also expressing high *Hox9* gene levels. The third classification group was represented by the heart, characterized by low *Hox* gene expression. This may reflect embryonic developmental processes, as *Hox* genes are known to exert minimal influence on heart formation, and are generally not expressed in the heart, except for the residual expression carried over by neural crest cells that invade the arterial pole of the heart and promote aorticopulmonary septation (*Kirby et al., 1983*). The great vessels were excluded from our sample collection, and therefore cells of neural crest origin were likely not captured in the analyses. The fourth classification group was represented by the kidney, which expressed intermediate to high levels of most anterior *Hox* genes up to *Hox11*, consistent with previous observations for the developing kidney (*Patterson and Potter, 2004*). The fifth category, represented by the mouse tail, had a posterior *Hox* code signature, represented by *Hox13*, which correlates with previous findings for human distal segment fibroblasts, represented by feet skin fibroblasts (*Rinn et al., 2006*; *Rinn et al., 2008*). Taken together with previous observations, these analyses confirm that adult organ fibroblasts retain positional *Hox* gene expression signatures, generally reflecting the embryological segmental identity of organ fibroblasts.

## Organ fibroblasts show unique molecular signatures

To highlight the unique transcriptomic signatures of these positionally distinct fibroblast pools, we performed a differential expression analysis and considered genes that were enriched by tenfold change or more in single organ fibroblasts relative to tail fibroblasts (*Figure 2*, *Figure 2—source data 1*). Gene ontology (GO) annotation revealed organ development programs; processes such as epithelial development, hepatoblast differentiation, lung lobe development, kidney development, reproductive process, and heart development were found enriched in their respective organ fibroblast pools (*Figure 2a*). Strikingly, signature embryonic transcription factors, that is, genes with established involvement in organ development, were enriched in organ-specific subsets, including *Tbx20*, a crucial transcription factor for heart development previously described in cardiac fibroblasts (*Furtado et al., 2014a*). Likewise, genes essential for lung morphogenesis (*Foxf1*) (*Costa et al., 2001*), liver development (*Hhex*) (*Keng et al., 2000*), early kidney formation (*Pax8*) (*Bouchard et al., 2002*), and gonad development (*Lhx9*) (*Birk et al., 2000*) were all specifically enriched in fibroblasts from their respective organs. Expression of signature genes was validated by quantitative polymerase chain reaction (qPCR) (*Figure 2b*, *Figure 2—figure supplement 1*) and immunocytochemistry (*Figure 3*, *Figure 3—figure supplement 1*). In general, signature gene expression patterns in embryonic fibroblasts were retained in fibroblasts from adult tissues (*Krt4*, *Krt6a*, *Serpinb5*, and *Hp* for skin; *Tbx20* and *Col2a1* for heart; *Foxf1* for lung; *Hhex* and *Foxa2* for liver, *Bmp7* and *Pax8* for kidney, *Cyp11a1* and *Lbx9* for gonad). Significant expression of *Hhex* and *Bmp7* were also found in several organs during embryonic development but were restricted to a single organ in adulthood. As an exception to single organ

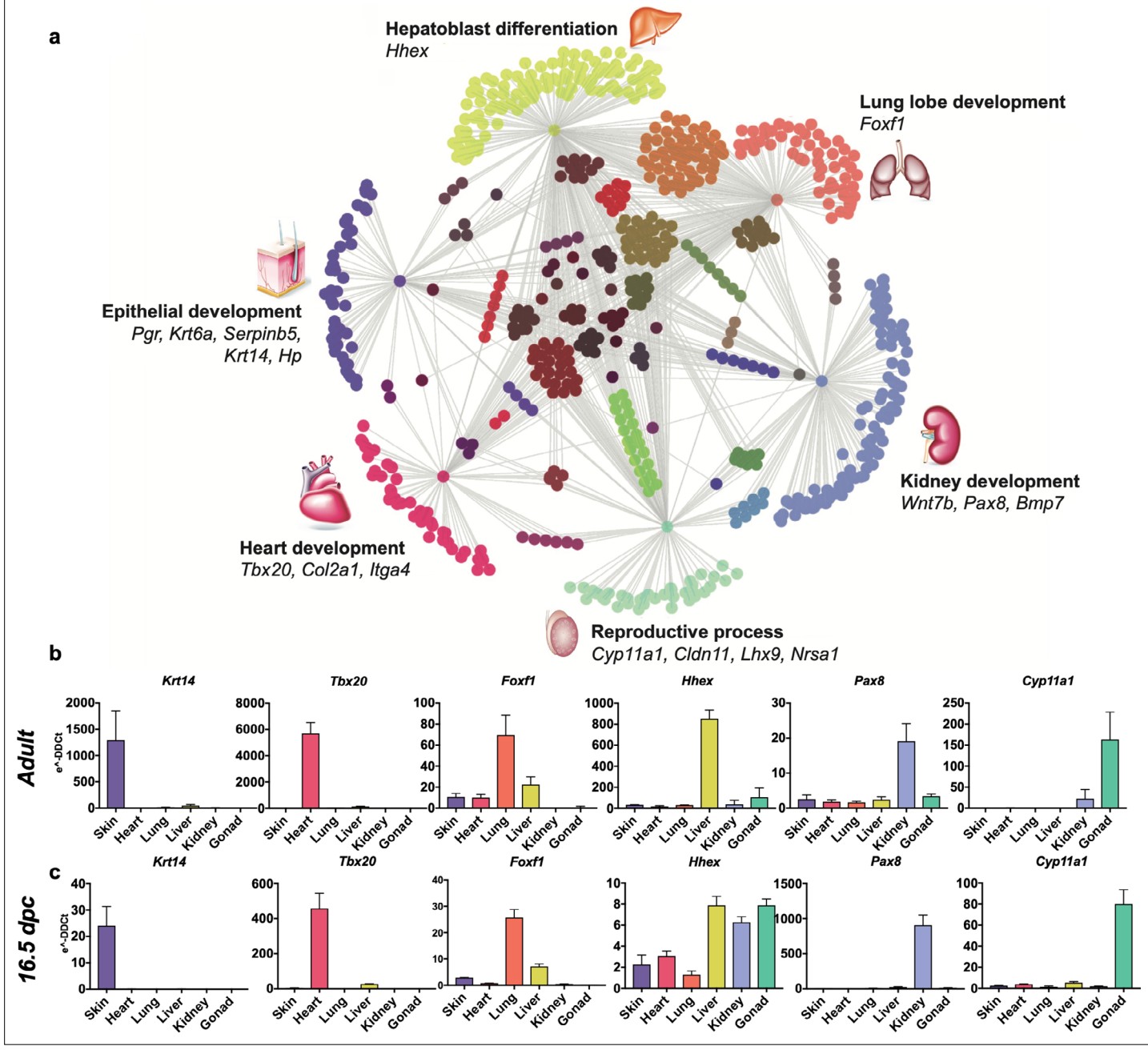

**Figure 2.** Embryological molecular signature of organ fibroblasts. (**a**) Cytoscape representation of network of genes (dots) singularly expressed in organ fibroblasts (only one gray edge between the gene and the organ) or shared among organs (multiple gray edges linking the gene to several organs). Genes involved in organ development are highlighted. (**b, c**) Validation of the expression of selected organ-enriched, developmental related genes using qPCR on cultured organ-derived fibroblasts isolated from adult mice (top row) or E16.5 embryos (bottom row). Data are mean ± standard error of the mean (SEM) of the $e^{-DDCt}$ values, on three technical replicates of merged biological samples. The housekeeping gene is *Hprt* and the reference sample is tail fibroblasts. Refer to *Figure 2—figure supplements 1–7*, *Figure 2—source data 1*.

The online version of this article includes the following source data and figure supplement(s) for figure 2:

**Source data 1.** Microarray data: expression of genes that were enriched by tenfold change or more in single organ fibroblasts compared to tail fibroblasts (*n* = 3).

**Figure supplement 1.** Embryological molecular signature of organ fibroblasts.

**Figure supplement 2.** Molecular signature of skin fibroblasts.

**Figure supplement 3.** Molecular signature of lung fibroblasts.

*Figure 2 continued on next page*

*Figure 2 continued*

**Figure supplement 4.** Molecular signature of liver fibroblasts.

**Figure supplement 5.** Molecular signature of kidney fibroblasts.

**Figure supplement 6.** Molecular signature of gonad fibroblasts.

**Figure supplement 7.** Molecular signature of cardiac fibroblasts.

**Figure supplement 7—source data 1.** Expression of cardiac fibroblast-enriched genes in human left ventricular biopsies from healthy and chronic ischemic heart failure patients.

enrichment, *Foxa2* was also substantially upregulated in lung fibroblasts (~20-fold), in addition to liver fibroblasts (~30-fold).

IPA analysis delineated top canonical pathways, diseases, functions, and networks associated with selectively enriched genes in each fibroblast population and supported the argument that fibroblasts retain molecular identity of their developmental organ of origin (*Figure 2—figure supplements 2–7*). Among organ-related processes enriched in fibroblast subsets were dermatological diseases and conditions and morphogenesis of the epithelial tissue for skin fibroblasts (*Figure 2—figure supplement 2*), respiratory system development for lung fibroblasts (*Figure 2—figure supplement 3*), liver development for liver fibroblasts (*Figure 2—figure supplement 4*), acute renal failure, metanephros development and kidney formation, and abnormal kidney development, disease and function for kidney fibroblasts (*Figure 2—figure supplement 5*), reproductive system development, function and disease, morphology of genital organs, and primary sex determination networks and reproductive system dysfunction for gonad fibroblasts (*Figure 2—figure supplement 6*), cardiovascular disease development and function, cardiac enlargement and disease, and cardiac developmental processes for heart fibroblasts (*Figure 2—figure supplement 7*).

To establish the translational relevance of our findings, the presence of 42 genes uniquely enriched in cardiac fibroblasts (log2, tenfold, False Discovery Rate (FDR) < 0.01) was determined in left ventricular heart biopsies from healthy (*N* = 5) and chronic ischemic heart failure patients (*N* = 5) (*Figure 2—figure supplement 7—source data 1*). Ischemic heart failure was chosen for analysis due to the likelihood of replacement fibrosis as a pathological signature. Out of the 42 murine cardiac fibroblast genes, 28 were present in both control and heart failure samples, including *Tbx20*, which was unchanged between control and heart failure. *FNDC1*, *FRZB*, *MFAP4*, and *OLFML1* were significantly upregulated in ischemic heart failure; while *MFSD2A*, *PNP*, and *SERPINA3N* were downregulated. Four genes had no human homolog and ten were not found in the human heart transcriptome dataset. These findings confirm general commonalities across species and further identify potential candidates of fibrotic cardiovascular disease interest for future investigation.

## Organ-enriched gene expression is retained at the single-cell level in freshly isolated and cultured fibroblasts

Single-cell RNAseq (scRNAseq) is a powerful tool to determine granularity of gene expression at the population level. To assess how organ signatures are reflected in freshly isolated fibroblasts, we reanalyzed the stromal cell dataset from a publicly available multiorgan scRNAseq study (the Mouse Cell Atlas) (*Han et al., 2018*). Focusing on lung, testis, kidney, liver, and neonatal heart cells, we unbiasedly identified eight populations, including three lung and two kidney subclusters (*Figure 4a*). Pairwise differential expression analysis supported a previously reported classification of lung fibroblasts populations (*Xie et al., 2018*), with two types of matrix fibroblasts – 'Lung A' (Col*14a1*, *Pi16*, *Dcn* enriched), 'LungB' (Col*13a1*, *Cxcl14*, *Tcf21* enriched) – and a group of myofibroblasts – 'LungC' (*Acta2*, *Myl9*, *Tagln* positive) (*Figure 4—figure supplement 1 a, b*). For kidney clusters, 'Kidney B' showed higher levels of canonical fibroblast markers *Dcn*, *Gsn*, and *Col1a2*, while the larger 'Kidney A' expressed relatively higher levels of genes involved in response to injury, or in renal carcinoma metastasis and progression (*Spp1* and *Krt8*), suggesting that this cluster is composed of tubular cells acquiring a mesenchymal phenotype in vivo (*Rudman-Melnick et al., 2020*; *Figure 4—figure supplement 1c, d*), as kidney epithelial cells are known to undergo dedifferentiation in vivo and in vitro to repair tubular injuries (*Kusaba et al., 2014*; *Van der Hauwaert et al., 2013*).

Overall, the expression of organ-specific (*Figure 4b*) and developmental related genes (*Figure 4c*) previously identified in the bulk cultured cell analysis was preserved, despite the reduced coverage

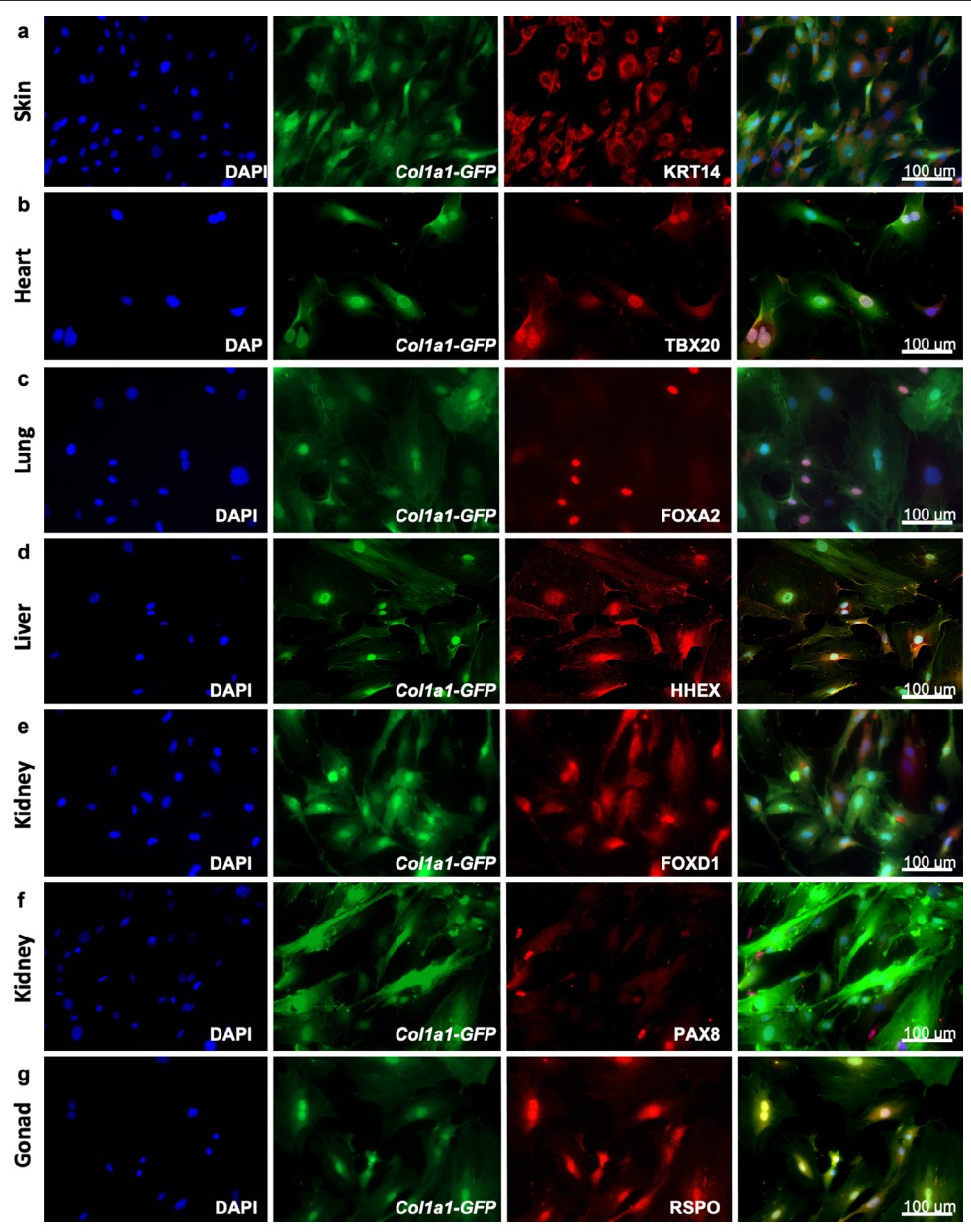

**Figure 3.** Adult fibroblasts express organ-specific transcription factors. Immunocytochemistry for development-related organ-enriched markers (KRT14, TBX20, FOXA2, HHEX, FOXD1, PAX8, and RSPO) on adult fibroblasts obtained from different organs of *Col1a1-GFP* mice and cultured for 5 days: (**a**) Skin fibroblasts stained with a mouse monoclonal anti-KRT14 antibody; (**b**) heart fibroblasts stained with a mouse monoclonal anti-TBX20 antibody; (**c**) lung fibroblasts stained with a rabbit monoclonal anti-FOXA2 antibody; (**d**) liver fibroblasts stained with a rabbit monoclonal anti-HHEX antibody; (**e**) kidney fibroblasts stained with a rabbit polyclonal anti-FOXD1 antibody; (**f**) kidney fibroblasts stained with a mouse monoclonal anti-PAX8 antibody; (**g**) testis fibroblasts stained with a goat polyclonal anti-RSPO1 antibody. As secondary antibodies, goat anti-mouse Alexa Fluor 568 was used for (**a**, **b**), (**f**); goat anti-rabbit Alexa Fluor 555 for (**c**, **d**) and donkey anti-goat Alexa Fluor 568 for (**f**). Representative images from three independent experiments on three biological replicates. Scale bar = 100 µm. Refer to *Figure 3—figure supplement 1*.

The online version of this article includes the following figure supplement(s) for figure 3:

**Figure supplement 1.** Adult fibroblasts express organ-specific transcription factors.

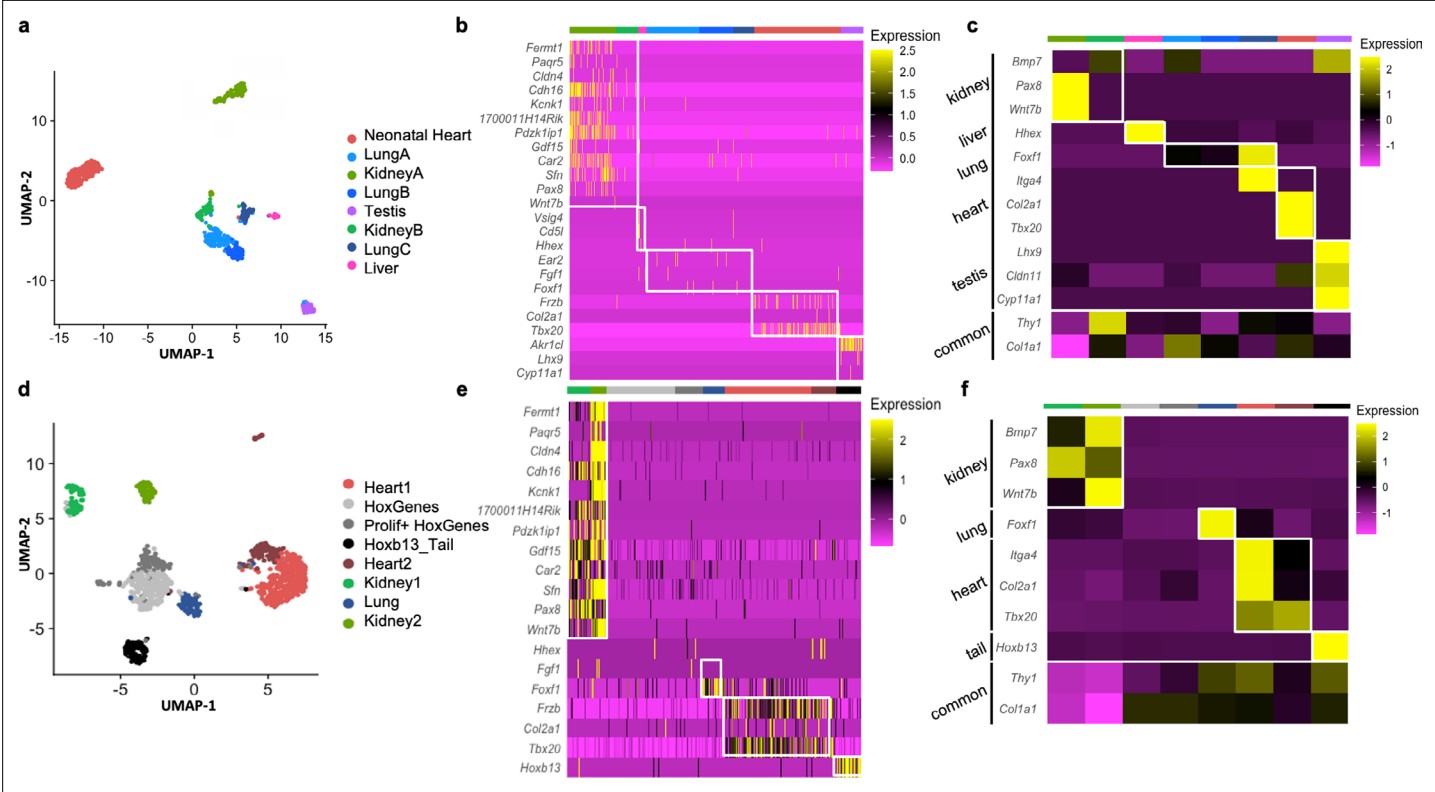

**Figure 4.** Analysis of organ-specific signatures at the single-cell level. (**a–c**) Reanalysis of the Mouse Cell Atlas stromal cell dataset. (**a**) Uniform Manifold Approximation and Projection (UMAP) visualization of selected stromal cell populations (682 cells). (**b**) Heatmap representing the expression of top-organ-specific genes, identified from the bulk RNAseq comparison, on individual cells in the single-cell RNAseq (scRNAseq) dataset. (**c**) Heatmap showing the average expression per population of organ development-related genes (same as shown in *Figure 3*). (**d–f**) scRNAseq analysis of mixed cultured stromal cells of different origins. (**d**) UMAP visualization of the captured cells. (**e**) Heatmap representing the expression of top-organ-specific genes, identified from the bulk RNAseq comparison, on individual cells. (**c**) Heatmap showing the average expression per population of organ development-related genes (same as shown in *Figure 3*). Refer to *Figure 4—figure supplements 1–2*, *Figure 4—source data 1*, and *Figure 4— source data 2*.

The online version of this article includes the following source data and figure supplement(s) for figure 4:

**Source data 1.** Analysis of the stromal cell aggregate from the Mouse Cell Atlas: markers genes per each population, and markers identified by pairwise comparison of the two kidney and three lung populations.

**Source data 2.** Analysis of in-house single-cell RNAseq (scRNAseq) data of merged cultured fibroblasts from different organs: markers genes per each population, and markers identified by pairwise comparison of the two kidney and two cardiac populations.

**Figure supplement 1.** Analysis of the organ-specific fibroblasts heterogeneity in freshly isolated cells at single-cell level.

**Figure supplement 2.** Analysis of the organ-specific fibroblast heterogeneity in cultured cells at single-cell level.

and expected heterogeneity at the single-cell level (*Figure 4b*). Interestingly, when multiple subclusters were present (as in lung and kidney fibroblasts) the expression of the organ-specific genes was enriched in myofibroblasts (Lung C expressed relatively higher levels of *Foxf1*) or activated fibroblasts (Kidney A expressed higher levels of *Pax8* and *Wnt7b*) (*Figure 4c*, *Figure 4—figure supplement 1*, *Figure 4—source data 1*).

Further confirmation of the organ-enriched program was obtained with scRNAseq of pooled primary cultures from different origins (kidney, liver, lung, heart, skin, testis, and tail) (*Figure 4d*). Unbiased clustering defined Kidney 1–2, Lung, Heart 1–2, and Tail fibroblasts. Two additional clusters were unclassified based on organ identity, although marked by the expression of Hoxc genes (*HoxGenes*) and proliferation genes (*Prolif + HoxGenes*) (*Figure 4d–f*, *Figure 4—source data 2*). Highly expressed (*Figure 4e*) and development-related genes (*Figure 4f*) from original bulk analysis were again confirmed in these organ populations. Both cultured kidney clusters (Kidney 1–2) expressed the epithelial stress response marker (*Spp1*) and were transcriptionally closer to freshly

isolated 'Kidney A' (*Figure 4—figure supplement 2c*), possibly representing two stages of tubular cells epithelial-to-mesenchymal transition (*Iwano et al., 2002*): Kidney1 had higher expression of myofibroblast genes (*Col4a1*, *Tagln*, *Myl9*, and *Sparc*) and the kidney-fibroblast-enriched gene *Pax8*; Kidney 2 strongly expressed epithelial genes (*Krt7, 8, 18, Epcam,* and *Clu*) (*Figure 4—figure supplement 2d, e*; *Figure 4—source data 2*). As for the cultured heart fibroblasts, Heart 1 displayed myofibroblast genes (*Acta2*, *Tagln*, and *Myl9*) and Heart 2 had enhanced signature of injury response/acutely activated fibroblasts (*Mt1*, *Ccl2*, *Clu*, and *Dcn*) (*Figure 4—figure supplement 2a-b*; *Figure 4—source data 2*; *Forte et al., 2020*). Overall, scRNAseq experiments showed that cultured cells present an activated/myofibroblast-like phenotype compared to freshly isolated cells and confirmed the retention of an organ-specific core transcriptome identity on both cultured and freshly isolated cells.

## Organ-enriched transcriptome is involved in the fibrotic response

To investigate the functional relevance of organ-enriched fibroblast transcriptomes, a CRISPR knock-down approach was used to downregulate core organ transcription factors, taking the heart as a model (*Figure 5*). ROSA^Cas9-GFP (*Platt et al., 2014*) adult cardiac fibroblasts were cotransfected with mCherry mRNA and GFP guide RNAs for determination of transfection and knockdown (KD) efficiency, respectively (*Figure 5a*). After 72 hr of transfection, mCherry was observed in roughly 67% of cells (*Figure 5b*) by flow cytometry, and GFP mRNA expression was downregulated by over 90% when compared with scrambled guides (negative control) (*Figure 5c*). GFP fluorescence was also dramatically decreased by 72 hr (*Figure 5a*; *Li et al., 2014*).

With this confirmation, *Gata4* and *Tbx20*, core transcription factors essential for heart formation in embryonic development (*Nim et al., 2015*), were knocked-down in cultured ROSA^Cas9-GFP cardiac fibroblasts, followed by bulk RNAseq analysis (*Figure 5d–j*). *Gata4* is expressed by all organ fibroblasts while fibroblast *Tbx20* expression is restricted to the heart. Despite similar KD efficiencies for both targets (~60%; *Figure 5d, e*), *Gata4* KD induced a higher number of dysregulated genes (red dots on volcano plots) compared with *Tbx20* KD. Among genes upregulated by tenfold in the original bulk organ analysis (eHF; *Figure 5f, g*), nine were dysregulated by *Gata4* KD and five by *Tbx20* KD. Only two of these genes were dysregulated in both conditions, including *Tbx20*, suggesting *Gata4* as a possible upstream regulator of *Tbx20*. A number of genes showed opposite regulation between *Gata4* KD and *Tbx20* KD (*Figure 5h*), confirming the specificity of KD response. These included cytokines and cytokine receptors (*Il11*, *Tnfsf18*, and *Ackr4*), genes involved in infection (*Ptgs2* and *Heyl*), cell adhesion and migration (*Spon2* and *Mmp10*). Of note, among the genes selectively upregulated in *Gata4* KD, *Il11* is a key mediator of organ fibrosis, possibly downstream of TGFβ (*Schafer et al., 2017*). Among genes upregulated in *Tbx20* KD, the downstream effector of Notch signaling *Heyl* is involved in cardiogenesis and is thought to repress *Gata4* expression (*Fischer et al., 2005*), *Mmp10* is upregulated in patients with end-stage heart failure (*Wei et al., 2011*), and it is involved in valve ossification (*Matilla et al., 2020*). KEGG pathway analyses (*Figure 5i*, *Figure 5—source data 1*) confirmed the involvement of *Gata4* and *Tbx20* in common but also diverse pathways. The top pathways uniquely upregulated by *Gata4* KD included Akt signaling, ECM–receptor interaction and renin secretion, implicating *Gata4* in the modulation of cardiac fibroblast growth and function (*Brilla et al., 1995*; *Li et al., 2015*; *Dai et al., 2019*). The top pathways upregulated by *Tbx20* KD involved IL-17 and relaxin signaling, as well as transcription misregulation in cancer. IL17 has been shown to regulate the fibrotic response in proinflammatory conditions such as psoriasis and pulmonary/liver fibrosis (*Lei et al., 2016*; *Zhang et al., 2019*; *Seki and Brenner, 2015*; *Blauvelt and Chiricozzi, 2018*), while relaxin has a well-established role in suppressing myofibroblast activation and ECM remodeling (*Ng et al., 2019*; *Martin et al., 2019*; *Samuel et al., 2017*).

In summary, both gene KDs affected matrix components and modulators (*Figure 5j*), as well as cell adhesion, cell–cell communication, and cell signaling genes. Markers of the epicardium, the external layer of the heart from which embryonic fibroblasts derive (*Quijada et al., 2020*), were also modulated, as well as several myocardial genes, found in low levels in cardiac fibroblasts and downregulated in *Gata4* KD. These results confirm the biological relevance of organ-specific fibroblast gene expression.

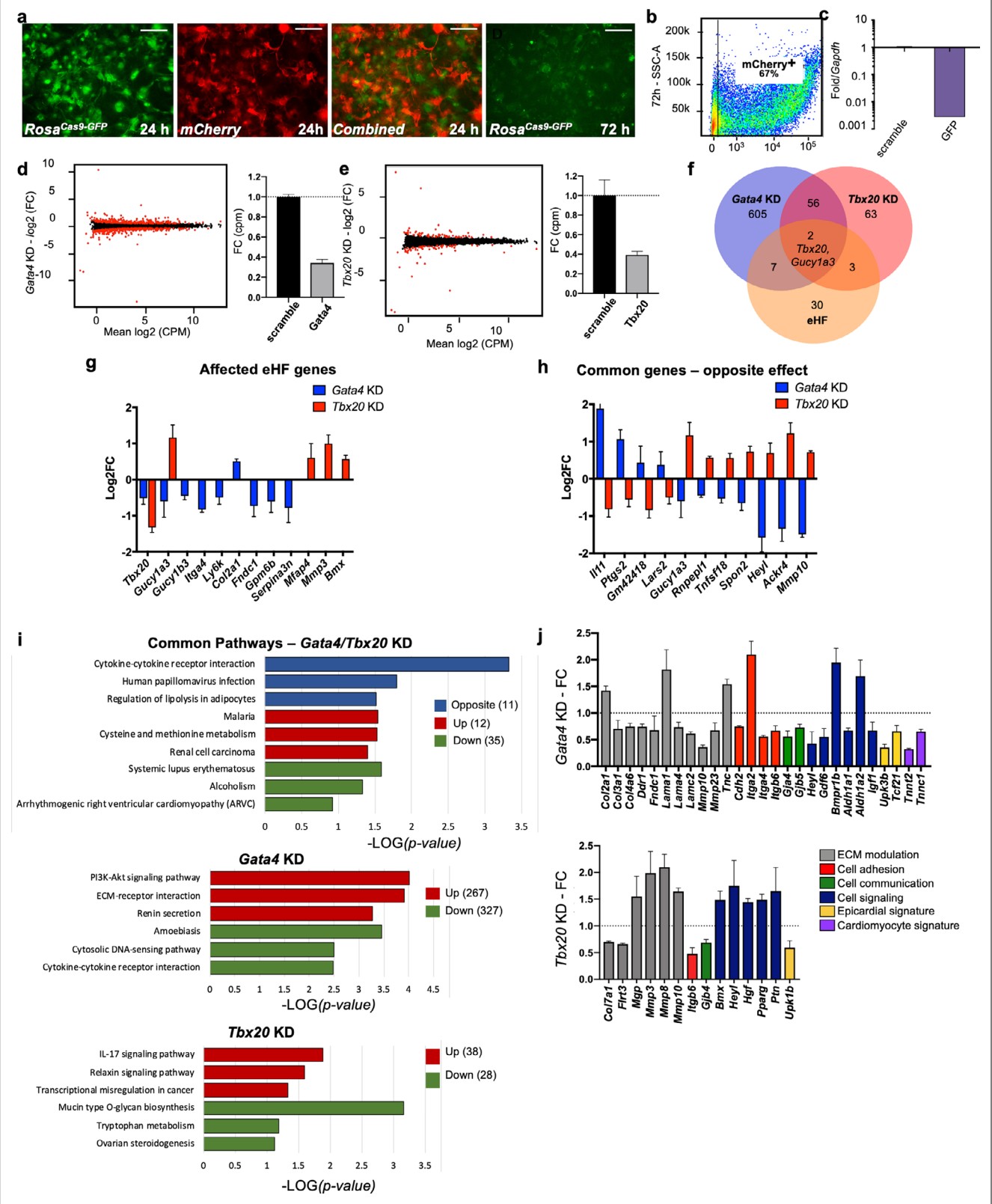

**Figure 5.** In vitro knockdown of core cardiac transcription factors. (**a**) Images of adult cardiac fibroblasts derived from *ROSA*<sup>Cas9-GFP</sup> mice, 24 and 72 hr post-transfection with two guide RNAs for *GFP* and CleanCap mCherry mRNA. (**b**) Flow cytometry plot showing the expression on mCherry in 67% of transfected cells (representative image of three independent experiments). (**c**) Relative quantification qPCR of *GFP*, normalized by *Gapdh* expression, in cells transfected with scrambled or GFP guide RNAs. (**d**, **e**) Volcano plots showing genes differentially expressed in *Gata4* (left) and *Tbx20* wns (right).

*Figure 5 continued on next page*

**Figure 5 continued**

Fold change in expression of *Gata4* (left) and *Tbx20* (bottom) in cells transfected with specific gRNAs versus scramble RNA, quantified through RNA sequencing. (**f**) Venn diagram showing the overlap among genes affected by Gata4 (*Gata4KD*) or Tbx20 (*Tbx20KD*) knockdown and number of genes upregulated by tenfold or more in heart fibroblasts (eHF) compared to other organs. (**g**) Plot showing changes in expression of eHF genes affected by *Gata4* (in blue) or *Tbx20* (in red) knockdown. (**h**) Genes regulated by both *Gata4* and *Tbx20* in opposing manner. (**i**) KEGG pathway analyses. Top panel: 58 genes affected in both *Tbx20* and *Gata4* knockdown; middle panel: 594 genes affected by *Gata4* knockdown; bottom panel: 66 genes affected by *Tbx20* knockdown. Blue – pathway changed in opposite directions, red – upregulated pathways, and green – downregulated pathways. (**j**) Hand-picked genes illustrate alterations in processes known to affect the cardiac fibrotic response for *Gata4* or *Tbx20* knockdown. All data are represented as fold changes over scrambled control (average ± standard error of the mean [SEM]; **d–j**) from bulk RNAseq of three biological replicates per condition. Selected significant genes have an FDR < 0.05. Refer to *Figure 5—source data 1*.

The online version of this article includes the following source data for figure 5:

**Source data 1.** CRISPR-Cas9 experiments: sequence of the guide RNAs; differentially expressed genes between Tbx20KD and Gata4 KD and corresponding controls.

## Organ fibroblast specificity affects tissue function in coculture systems

The studies described above confirmed that fibroblasts retain an organ-specific transcriptome from embryonic development to adulthood, and that their identities are largely maintained in cultured cells, suggesting that fibroblast transcriptomes may be important for in vivo organ function. To determine

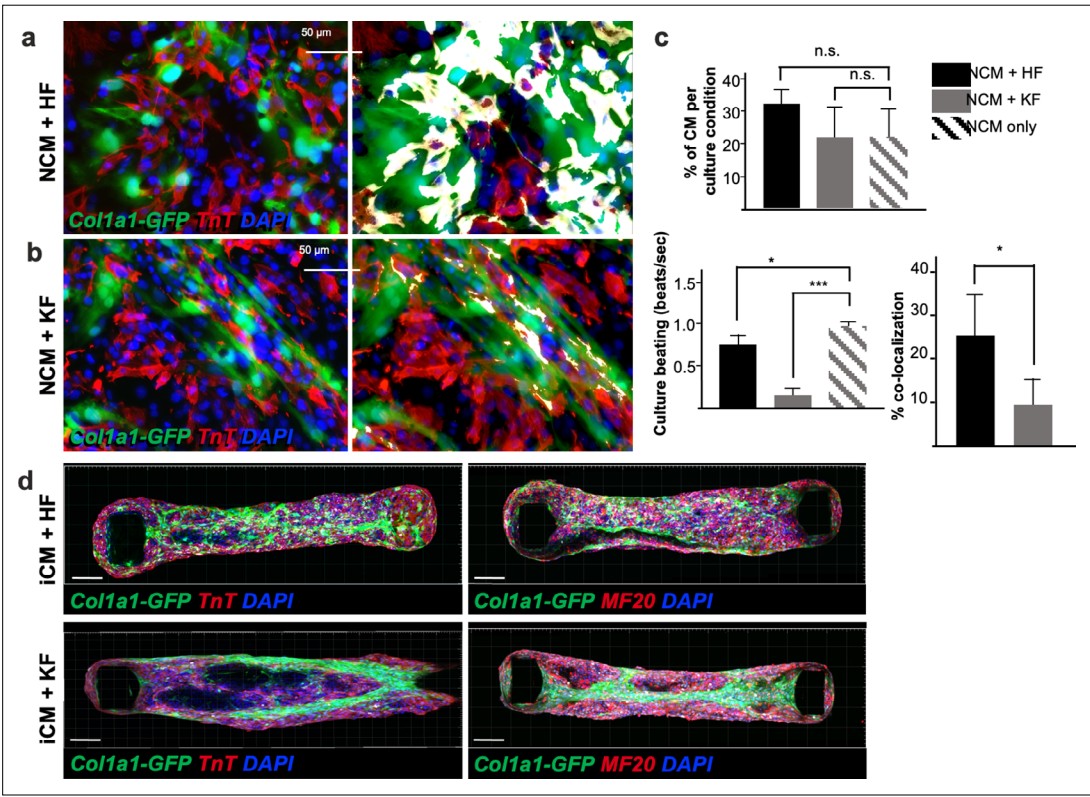

**Figure 6.** Adult fibroblasts retain tissue-specific function in vitro: impaired neonatal cardiomyocyte beating in presence of kidney-derived fibroblasts. Immunocytochemistry for TnT on 2D coculture of neonatal ventricular cardiomyocytes (NCM, TnT+ in red) with either adult cardiac fibroblasts (HF) in (**a**) or kidney fibroblasts (KF) in (**b**), isolated from *Col1a1-eGFP* mice (in green). Nuclei are labeled with 4',6-diamidino-2-phenylindole (DAPI) (in blue). The right panel shows colocalization of the two cell types (green+ and red+) in white. (**c**) Quantifications of the percentage of cardiomyocytes per culture condition (top), beating of the 2D cultures expressed in beats per second (bottom left) and percentage of colocalization (bottom right). (**d**) Confocal Z-stack images reconstructed with Imaris, of cardiac microtissues constituted of 85% hiPSCs derived cardiomyocytes (iCM) and 15% of either cardiac (HF) or kidney (KF) fibroblasts, stained for TnT (in red, left panels) or MF20 (in red right panels). The adult fibroblasts were isolated from *Col1a1-eGFP* mice (in green), nuclei stained with DAPI (in red). Forty-eight organoids were generated per each fibroblasts cell type. Scale bar = 50 μm. All data in (**c**) are mean ± standard error of the mean (SEM) on three independent experiments, p values were calculated by two-sample *t*-test; * = p<0.05, *** = p<0.005 Refer to *Video 1* and *Video 2*.

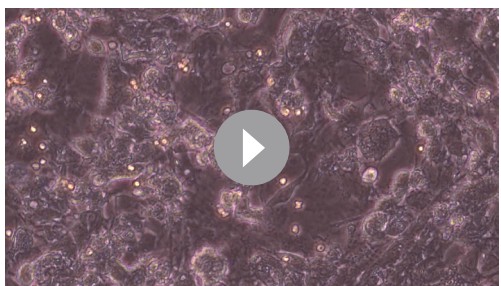

**Video 1.** Coculture of adult cardiac fibroblasts with neonatal ventricular cardiomyocytes. ×20 magnification. https://elifesciences.org/articles/71008/figures#video1

whether the source of organ fibroblast affects organ function, 2D and 3D cocultures of cardiomyocytes (CMs) with adult kidney and cardiac fibroblasts were performed (*Figure 6*).

For 2D cultures, neonatal ventricular CMs were plated with *Col1a1-GFP*+ fibroblasts isolated from the adult kidney or heart (*Kanisicak et al., 2016*). Within 24 hr, coculture with adult kidney fibroblasts almost completely impaired CM contractility (*Figure 6b, c*, *Video 1*), although the number of CMs present in both cultures was not significantly different. Conversely, cardiac fibroblast coculture resulted in a syncytium of cells beating in synchronism (*Video 2*) at a relatively lower pace than neonatal CMs alone (*Figure 6c*), possibly reflecting an effect of adult HF on neonatal CM maturation (*Wang et al., 2020*). In addition, cardiac fibroblasts were well integrated with neonatal CMs, as shown by the percentage of colocalization, while kidney fibroblasts and CMs seemed to repel each other. To confirm these findings, 3D cardiac microtissues we generated, as previously described (*Boudou et al., 2012*; *Hinson et al., 2015*). A suspension of 85% human-induced pluripotent stem cell-derived CMs (iCMs) and 15% adult cardiac or kidney fibroblasts was loaded on millitissue devices with pairs of cantilevers to generate force. As expected, cardiac fibroblasts were homogeneously interspersed, while kidney fibroblasts were aggregated to the center or periphery of the organoids (*Figure 6d*), indicating lack of integration between the two cell types. These results implicate organ-specific fibroblasts in imparting their cognate tissue integrity.

## Ectopically transplanted fibroblasts retain core transcriptional identity

To investigate if adult fibroblasts maintain their organ-specific signature when exposed to a different tissue microenvironment in vivo, *ROSA*^mT/mG fibroblasts from tail, heart, and kidney were absorbed on surgical gel foam and transplanted under the kidney capsule of syngeneic C57BL6/J mice (*Figure 7a*). Three days post-transplantation, kidneys were dissected and sorted (*Figure 7b*) to determine transcriptional changes in transplanted fibroblasts. Transplanted heart (HFs), kidney (KFs), tail (TFs) fibroblasts and corresponding in vitro cultured controls (HFc, KFc, TFc) were processed for bulk RNAseq. Multidimensional scaling plot for all samples showed that the three organ fibroblast types retained a distinct identity post-transplant, despite a reduced transcriptomic separation (*Figure 7c*).

To assess eventual changes in organ-specific identity, we compared the expression of sorted and cultured heart and kidney fibroblasts relative to tail fibroblasts, and we analyzed the expression of heart-enriched (eHF) and kidney-enriched (eKF) genes identified from the initial bulk RNA analyses (*Figure 2—source data 1*, *Figure 2—figure supplements 2–7*). We observed that fibroblasts generally maintained their core identity after transplant. Of the 26 genes enriched in HFc, 21 (80.7%) were similarly modulated in HFs, only two were downregulated and three were not detected (*Figure 7d*). As expected, eHF gene expression was low or downregulated in KFc compared to TFc and kept a similar

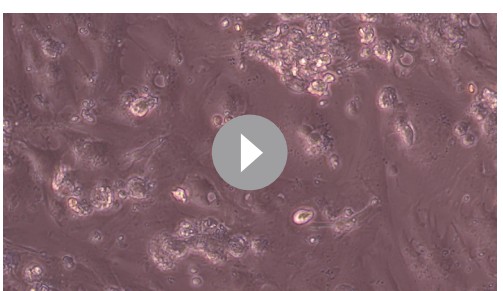

**Video 2.** Coculture of adult kidney fibroblasts with neonatal ventricular cardiomyocytes. ×20 magnification. https://elifesciences.org/articles/71008/figures#video2

expression pattern post-transplant (*Figure 7d*). Of the 47 eKF significantly expressed in KFc, 41 (87.2%) were modulated in the same direction in KFs, one gene was downregulated, and five genes were not detected. Twenty-six (55.3%) eKF genes were also found in HFc, 17 mildly upregulated and 9 downregulated. Of these, 14 were similarly regulated in HFs, 1 gene was not detected and 11 were differentially regulated (7 upregulated, 4 downregulated). An additional 13 eKF genes were detected in HFs, all mildly upregulated except for one downregulated, showing an adaptation to the new microenvironment (*Figure 7e*).

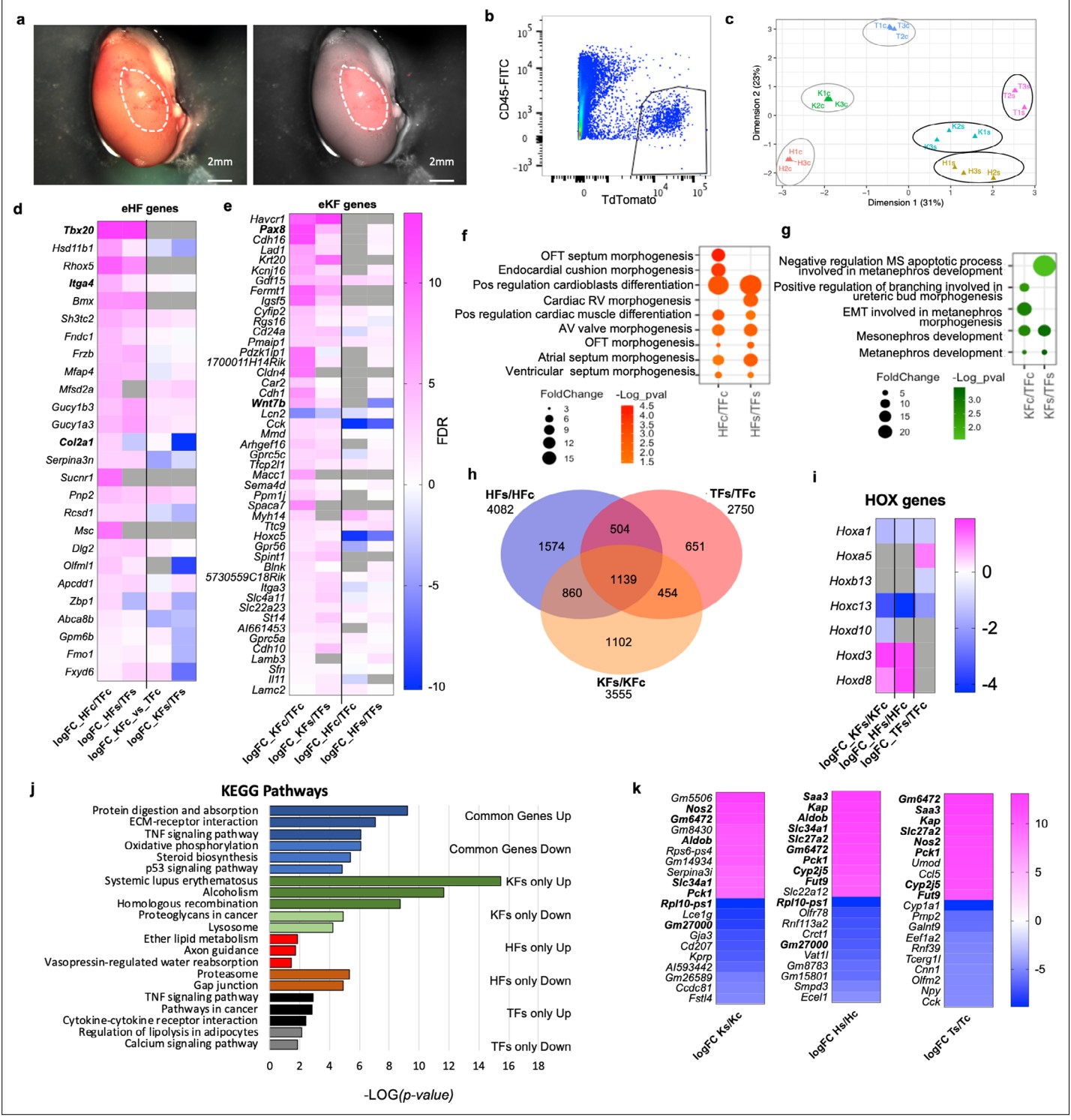

**Figure 7.** Fibroblast tissue-specific response to in vivo ectopic transplantation under the kidney capsule. (**a**) Representative images of a dissected kidney to highlight the area where TdTomato+ cells were transplanted; brightfield image on the left; brightfield overlaid with the fluorescence image acquired in the red channel on the right. (**b**) Flow cytometry plot showing the gating strategy used to isolate live CD45−;TdTomato+ cells 3 days post-transplantation in the kidney capsule. (**c**) Multidimensional scaling plot calculated on the top 500 genes postnormalization to visualize the transcriptomic similarity among all samples. (**d–h**) Comparison of HF and KF gene expression to TF in culture (HFc/TFc, KFc/KFc) and post-transplant in the kidney capsule (HFs/TFs, KFs/KFs). (**d**) Heatmap showing expression of significantly regulated eHF genes in the four conditions. (**e**) Heatmap showing the expression of eKF genes in the four conditions. (**f**) Dot plot indicating GO terms associated with cardiac development, identified from the DAVID database analysis of HFs or HFc-enriched genes. (**g**) Dot plot indicating the GO terms associated with cardiac development, identified from the DAVID

*Figure 7 continued on next page*

*Figure 7 continued*

database analysis of KFs or KFc-enriched genes. (**h–k**) Analysis of the differential expression by experimental condition: transplanted cells versus cells in culture HFs/HFc, TFs/TFc, and KFs/KFc. (**h**) Venn diagram showing the significant differentially regulated genes (FDR < 0.05) in the three comparison sets. Only genes with a logFC >1 or <−1 were considered. (**i**) Heatmap of differentially regulated *Hox* genes. (**j**) Bar plot of the KEGG pathway analysis on the common regulated genes (light blue – downregulated, dark blue – upregulated), and genes uniquely modulated in KF (dark green – upregulated, light green – downregulated), HF (red – upregulated, brown – downregulated), and TF (black – upregulated, gray – downregulated). (**k**) Heatmaps showing the top 10 upregulated and top 10 downregulated genes for each dataset. In bold are the genes shared in two different sets of comparisons. Data are means of three biological replicates per condition. All the heatmaps (**d, e, j, k**) The average log fold change. For (**d, e**) Genes were ordered based on the FDR (smaller to larger value) of the comparison in the first column, HFc/TFc and KFc/TFc, respectively. Dot plots (**f, g**) and bar plot (**j**) data were organized based on the log transformation of the p values (−log(p value)). eKF – kidney-fibroblast-enriched genes, same as shown in *Figure 2—figure supplement 5*, eHF – heart-fibroblast-enriched genes, same as shown in *Figure 2—figure supplement 7*, KFs – transplanted kidney fibroblasts, HFs – transplanted heart fibroblasts, TFs – transplanted tail fibroblasts, KFc – kidney fibroblasts in culture, HFc – heart fibroblasts in culture, and TFc – tail fibroblasts in culture. Refer to *Figure 7—figure supplement 1*.

The online version of this article includes the following figure supplement(s) for figure 7:

**Figure supplement 1.** Analysis of organ fibroblast-specific responses to transplant under the kidney capsule.

GO analysis of KF- or HF-enriched genes in culture or post-transplant using *DAVID* (Database for Annotation, Visualization and Integrated Discovery, *Dennis et al., 2003*) revealed terms related to organ development, in line with the previous observations (*Figure 7f, g*). The top GO terms for both HFc and HFs were related to cardiac morphogenesis and cardioblast differentiation; the top terms for KFc and KFs were related to mesonephros and metanephros development. In summary, both HFs and KFs maintained their core transcriptomic identity compared to TFs.

## Ectopically transplanted fibroblasts adapt to a new microenvironment

To analyze the differential HF, KF, and TF responses to the transplantation, gene expression of each post-transplant fibroblast type was compared with its equivalent control kept in culture. Despite the retention of organ identity signatures, numerous genes were modulated in transplanted fibroblasts compared to cultured controls (4082 genes for HFs/HFc, 3555 for KFs/KFc, and 2750 for TFs/TFc) (*Figure 7h*). These included 4–5 *Hox* genes per cell type, showing a modulation of the cell-type-specific positional code. *Hoxa1* (the highest expressed in HFc) and *Hoxc13* were downregulated in all conditions (*Figure 7i*). The tail-enriched *Hoxb13* was downregulated in TFs, while *Hoxa5* was increased. *Hoxd3* and *Hoxd8* were upregulated in both HFs and KFs; and *Hoxd10* was downregulated in KFs. Interestingly, *Hoxd8* is important for the maintenance of epithelial phenotype in adult kidney and is expressed in the ureteric bud during development (*Di-Poï et al., 2007*), while *Hoxd10* is diffusely expressed in kidney mesenchyme in embryos; both *Hoxd10* and *Hoxd3* regulate *Itga3* expression and have been involved in different types of cancer (*Hua et al., 2020*). The differential regulation of *Hoxd10* and *Hoxd3* may suggest the acquisition of a cortex-like phenotype by transplanted KFs.

KEGG analysis of the genes modulated in response to the transplant showed upregulation of pathways related to fibrosis and damage response (TNF signaling, ECM–receptor interaction, protein digestion, and absorption) and downregulation of oxidative phosphorylation, steroid biosynthesis, and p53 signaling among the common regulated genes (*Figure 7l*, *Figure 7—figure supplement 1a*). Genes uniquely upregulated in KFs/KFc were related to homologous recombination and cell cycle, with pathways including several histone genes (Alcoholism, Systemic lupus erythematosus, *Figure 7—figure supplement 1a*); those selectively upregulated in HFs/HFc were associated with cell migration (axon guidance), vasopressin regulated water absorption and lipid metabolism; and to proinflammatory pathways for TFs/TFc (TNF signaling, cytokine–receptor interactions, pathways in cancers) (*Figure 7l*). Similarly, IPA analysis revealed that the most significantly affected canonical pathways were related to fibrosis (hepatic fibrosis signaling, GP6 signaling); cell migration (Axonal Guidance Signaling); acute phase response, inflammation, and cholesterol biosynthesis (*Figure 7—figure supplement 1b*). Interestingly, Cardiac Hypertrophy Signaling (including profibrotic signals AngII, TGFβ, and IGF1) and HIF1a signaling were predicted to be downregulated in KFs and upregulated in HFs and TFs, possibly inferring a better resilience of KFs to the kidney capsule environment.

Among the top 10 upregulated genes in transplanted fibroblasts, 6 were shared by the ectopically transplanted HFs and TFs, including the serum amyloid A *Saa3*, secreted during the acute phase of inflammation (*Forte et al., 2020*); two metabolic enzymes, *Slc27a2*, primarily expressed in kidney and

liver involved in lipid biosynthesis and fatty acid degradation, and *Pck1* key regulator of gluconeogenesis; the cytochrome gene *Cy2j5* involved in vasorelaxation (*Agba et al., 2020*); the kidney abundant protein *Kap*, androgen-regulated, proximal tubule-specific not expressed at detectable levels in tissue other than the kidney (*Toole et al., 1979*), *Fut9* a fucosyltransferase with the highest expression in adult pancreas, placenta, kidney (*Figure 7k*). In summary, while transplanted fibroblasts maintained their core identity, they responded to the kidney microenvironment by expressing a subset of kidney-specific genes, modulating positional code genes and activating common and cell-specific pathways in the attempt to adapt to the new, more hypoxic condition.

## Discussion

The ability to target-specific organ fibroblasts has long been impaired by the misconception that fibroblasts were functionally and phenotypically homogeneous cells, deputized to synthesizing and organizing the ECM, an idea possibly fostered by a common embryonic origin in the primary mesenchyme (*LeBleu and Neilson, 2020*). However, recent advances in lineage tracing and single-cell transcriptomic have revealed an extensive intra- and interorgan heterogeneity (*Henderson et al., 2020*; *LeBleu and Neilson, 2020*). Multiorgan studies show that B cells (*Rocha-Resende et al., 2020*), endothelial cells (*Paik et al., 2020*), fibroblasts (*Muhl et al., 2020*; *Han et al., 2018*), and transcriptomes tend to cluster separately based on the organ of origin, suggesting a strong influence of the anatomical location or the local microenvironment on the cell state. A recent study has shown that, despite the likely tissue-specific imprinting, fibroblast subclusters across multiple organs present a common hierarchy with two universal subtypes, with distinct localization within the tissue (Col15a1+ – parenchymal; Pi16+ – adventitial), which in turn generate other more specialized or activated fibroblasts states (*Buechler et al., 2021*). While our study did not prove or disprove this hypothesis, we were not able to unbiasedly classify our cell subtypes based on this principle. However, our data are in agreement with abovementioned studies that identified subclusters based on organ of origin.

We previously reported that fibroblasts isolated from the adult mouse heart retain a cardiogenic transcriptional program (*Furtado et al., 2014a*). Here, we compared primary cultures of fibroblasts isolated from organs of different anatomical positions to expand our previous analysis and assess whether development-related genes contribute to the fibroblast interorgan functional heterogeneity. The results of this analysis highlight the presence of an organ-enriched positional code, and the expression of core genes that represent the developmental signature of fibroblast organ origin previously thought to be restricted to the parenchymal component. These molecular profiles are established during embryogenesis, consistent with the fact that organ fibroblasts are not generated from a common progenitor pool but arise independently in different body segments and organs during embryonic development and persist to adulthood.

As in our previous study (*Furtado et al., 2014a*), we chose to analyze cultured fibroblasts to reduce the risk of contamination from parenchymal cell mRNA. Fibroblast expression patterns in culture were recapitulated in freshly isolated single cells, mostly enriched in activated or myofibroblast-like fibroblast subclusters. These gene signatures can predict the tissue of origin of a mixed population of primary cultured cells analyzed at the single-cell level. Using the heart as a model, we show that signature genes contribute to organ fibroblast function, as evidenced by the deregulation of several profibrotic and proinflammatory genes with knockdown of core transcription factors *Gata4* (expressed in all fibroblast types) and *Tbx20* (cardiac specific) in cultured adult cardiac fibroblasts. These results place *Gata4* upstream of *Tbx20,* both of which upregulate distinct profibrotic signals, modulate genes involved in extracellular modulation and cell adhesion, and have opposite effects on cytokine–cytokine receptor expression, confirming that the core cardiogenic program in cardiac fibroblasts is involved in regulating their function.

Dermal fibroblasts from different sites of the body have shown different efficiency of reprogramming into induced pluripotent stem cells (*Sacco et al., 2019*), but not much is known about other fibroblast tissue-specific functions. The coculture studies presented here further reinforce the importance of fibroblast core transcriptomes for specialized organ function: while interspersion of cardiac fibroblasts within CM cultures facilitated the propagation of the electric pulse forming a syncytium, cocultured kidney fibroblasts clustered separately and inhibited CM contraction, both in 2D and 3D assays. These findings carry repercussions to in silico organ bioengineering, where combining the correct match of diverse organ cell types may be essential for proper organ formation. Indeed,

human-induced pluripotent stem cell-derived cardiac stromal cells enhance maturation of cardiac microtissues (*Giacomelli et al., 2020*). In addition, the source and type of organ scaffolding, mainly deployed by fibroblasts, is essential for the recreation of organs in a dish (*Taylor et al., 2017*).

Previous studies have shown that skin fibroblasts and mesenchymal cells from different organs keep a positional identity (*Rinn et al., 2006*; *Rinn et al., 2008*; *Ackema and Charité, 2008*). For mesenchymal cells, the *Hox* code was maintained also in culture, although whether this depended on cell-to-cell contact remained to be determined (*Ackema and Charité, 2008*). Here, we show that adult fibroblasts, isolated from a variety of organs, preserve the expression of *Hox* genes in culture, but that tissue-specific *Hox* genes were downregulated after ectopic transplantation under the kidney capsule, suggesting that cellular environment can induce reprogramming of positional codes. Interestingly, control kidney fibroblasts also presented changes in the *Hox* code after transplantation, possibly reflecting the adaptation to the space between the capsule and the cortex, with the decrease of the mesenchyme gene *Hoxd10* and increase of *Hoxd3* important for the maintenance of an epithelial phenotype in adult kidney. All three transplanted fibroblast types (heart, tail, and kidney) presented an activated phenotype, involving initiation of the acute phase response, profibrotic signals, and metabolic changes. While transplanted heart and tail fibroblasts showed a clearer activation of proinflammatory pathways, genes associated with cell migration, and HIF1a signaling; transplanted kidney fibroblasts appeared more resilient in their native milieu, with the activation of pathways related to proliferation and upregulation of genes indicative of a more epithelial cortex-like phenotype. Both heart and kidney fibroblasts retained a 'memory' of their organ of origin, defined by the resilient expression of the core of development-related genes when compared to tail fibroblast control. It remains to be determined if this memory can be erased by a longer residence in the ectopic microenvironment. In light of recent studies on endothelial cells (*Yucel et al., 2020*; *Jambusaria et al., 2020*), we propose that the expression of organ-specific genes, previously thought to be restricted to parenchymal cells, may form the basis for organ cohesiveness and performance.

In summary, adult fibroblasts maintain a lasting blueprint of the organ in which they reside, reflective of their developmental origin, which likely plays a role in the orchestration of the tissue-specific homeostasis and reparative response. Exploiting the organ-specific properties of fibroblasts may be a valuable strategy for the targeted control of organ fibrosis, an integral feature of organ failure and disease progression affecting a multitude of pathologies.

## Materials and methods

### Mice

All experiments were performed with young adult (8–12 weeks old) C57BL/6J, *Gt(ROSA)26Sor^tm1.1(CAG-cas9*,-EGFP)Fezh*/J (*ROSA^Cas9-EGFP*), *ROSA^mt/mg* (JAX Stock# 007576)(*Muzumdar et al., 2007*), *Col1a1-GFP* (*Yata, 2003*) male mice and *Col1a1-GFP* E16.5 embryos. All animal experimentation conformed with local (Jackson Laboratory) and national (NHMRC and NIH) guidelines.

### Fibroblast isolation and sorting

Liver, heart, lung, kidney, tail, gonad, and ventral skin of adult mice and E16.5 embryos were dissected and finely minced. Fibroblasts were isolated using enzymatic digestion with 0.05% Trypsin/Ethylenediaminetetraacetic acid (EDTA)(Gibco) under agitation at 37°C for 30–40 min. Cells were spun and plated in 10 cm² dishes and cultured to semiconfluence in Dulbecco's Modified Eagle Medium (DMEM) (Thermo Fisher) high glucose supplemented with 10% Fetal Bovine Serum (FBS) (Thermo Fisher), 1% sodium pyruvate (Thermo Fisher), 1% penicillin–streptomycin (10,000 U/ml) (Thermo Fisher), 1% GlutaMAX Supplement (Thermo Fisher) in a 5% $CO_2$ incubator at 37°C. Passage 0 cells were then trypsinized using TrypLE (Thermo Fisher) and further processed for flow cytometry, labeled using CD90-AF647 (BioLegend), CD45-PeCy7, and CD31-Pe (eBioscience) in 2% FBS in Hank's balanced salt solution (HBSS) (Thermo Fisher) and sorted using Influx or Aria II Sorter (BD). The CD90+; CD45−; CD31− fraction was collected for mRNA isolation (*Figure 1—figure supplement 1*). Adult fibroblasts from *ROSA^Cas9-EGFP* and *Col1a1-GFP* were sorted using CD90-APC (BioLegend), CD45-APCCy7(BioLegend), and CD31-PECy7 (BD) after 3 or 5 days, respectively.

## Microarray assay

Sorted organ fibroblasts were resuspended in Cell Lysis buffer, further processed for total RNA isolation using the RNAqueous Micro kit (Thermo Fisher) and DNAse digested on column. Fibroblasts from individual mice were used for each replicate. Triplicates or more were used for each organ. Samples were further processed by the Monash Health Translational Precinct Medical Genomics Facility and ran on Agilent SurePrint G3 mouse gene expression arrays (single color).

## Bulk RNA sequencing

Total RNA was isolated from heart tissue using miRNeasy Mini kit (Qiagen); from cultured fibroblasts (CRISPR experiment) and sorted fibroblasts (kidney capsule experiment), using RNeasy Micro kit (Qiagen) according to the manufacturer instruction and including the optional DNase digest step. Sample concentration and quality were assessed using the Nanodrop 2000 spectrophotometer (Thermo Scientific) and the Total RNA Nano or Pico assays (Agilent Technologies).

For human heart samples, libraries were constructed using the KAPA RNA Hyper Prep Kit with RiboErase (HMR) (KAPA Biosystems), according to the manufacturer's instructions. For cultured fibroblasts (CRISPR experiment), libraries were constructed using the KAPA mRNA HyperPrep Kit (KAPA Biosystems), selecting polyA containing mRNA using oligo-dT magnetic beads, according to the manufacturer's instructions. For cells isolated from the kidney capsule, given the low RNA input, libraries were constructed using the SMARTer Stranded Total RNA-Seq Kit v2-Pico (Takara), according to the manufacturer's protocol.

All the libraries were checked for quality and concentration using the D5000 ScreenTape assay (Agilent Technologies) and quantitative PCR (KAPA Biosystems), according to the manufacturers' instructions; pooled and sequenced 75 bp paired-ended (human samples) or single-end (cultured and sorted fibroblasts) on the NextSeq 500 (Illumina).

## Single-cell RNA sequencing

Fibroblasts isolated from the different tissues were FAC sorted and loaded onto a single channel of the 10X Genomics Chromium single-cell platform. Briefly, cells were loaded for capture using the v2 single-cell reagent kit. Following capture and lysis, cDNA was synthesized and amplified (14 cycles) as per manufacturer's protocol (10X Genomics). The amplified cDNA was used to construct an Illumina sequencing library and sequenced on a single lane of a HiSeq 4000.

## Bioinformatics analyses

For microarray experiments, data extraction and preprocessing were performed as described previously (*Furtado et al., 2014b*). In brief, raw single-channel signals were extracted (Agilent Feature Extraction Software v.11.0.1.1), and quality control was performed using the default 'Compromised' option in (GeneSpring GX v.12.6), with threshold raw signal of 1.0. The approximate mean of 24 samples × ~55,000 probes (10,000) was used as a natural threshold between high- and low-intensity probes. If several probes represented a single gene, the mean of these probes was used. Probes that could not be mapped to any gene were discarded. Log2 transformation and quantile normalization were done using the R package *limma* v3.48.3. Differential analysis was performed using *limma* v3.48.3, which fits a linear model to the gene expression data, revealing the differential expression patterns (Benjamini–Hochberg adjusted p value <0.05 and fold change >2). These genes were extracted from the transcriptome to generate a heatmap together with hierarchical clustering dendrograms using Multi-Experiment Viewer (MeV) (*Saeed et al., 2003*). Differentially expressed genes showing more than tenfold change in any given organ were retrieved and an interaction file listing in which organs these genes were enriched was constructed. The interaction file was used as input for Cytoscape (*Shannon et al., 2003*) in order to reconstruct the network of genes shared by two or more organs, or specifically enriched in only one organ. The network layout was constructed using a Spring Embedded layout and MeV. GO over-representations for the organ-specific subset of genes was performed using the Cytoscape Bingo plug-in. Ingenuity Pathway Analysis was performed using the IPA software (Qiagen).

For single-cell RNA sequencing analyses of freshly isolated fibroblasts, stromal cell data from the Mouse Cell Atlas (*Han et al., 2018*) were kindly provided by Dr. Guoji Guo and Dr. Huiyu Sun. The data were reanalyzed using Seurat v3 (*Stuart et al., 2019*). Cells with less than 200 and more than 2500 transcripts were filtered out. Out of the original aggregate, containing 21 samples and 4830

cells, 5 populations of interest were selected for further analysis: 'Lung', 'Testis', 'Kidney', 'Liver', and 'NeonatalHeart', corresponding to 682 cells. Data were natural-log normalized and scaled using the top-2000 most variable features in the raw data. Principal component analysis (PCA) dimensionality reduction was calculated on 50 principal components; the Uniform Manifold Approximation and Projection (UMAP) dimensional reduction was calculated on 24 dimensions; cluster determination was performed using shared nearest neighbor at a 0.5 resolution. Cluster marker genes were identified with the *FindAllMarkers* function, using the default Wilcoxon Rank Sum test, at a threshold of 0.25 and a minimum difference in the fraction of detection (*min.diff.pct*) of 0.3. Pairwise comparisons were done using the *FindMarkers* function, with MAST assay and only testing genes that are detected in 25% of cells in either of the two populations (*min.pct* = 0.25).

For bulk RNAseq analysis on cultured fibroblasts post-CRISPR-Cas9 knockdown or kidney capsule implant: Single-end, Illumina-sequenced stranded RNA-Seq reads were filtered and trimmed for quality scores >30 using a custom python script. The filtered reads were aligned to *Mus musculus* GRCm38 using RSEM (v1.2.12) which performed alignment using Bowtie2 (v2.2.0) (command: rsem-calculate-expression -p 12 `--phred33-quals --seed-length` 25 `--forward-prob` 0 `--time --output-genome-bam-- bowtie2`). RSEM calculated expected counts and transcript per million. The expected counts values from RSEM were used in the edgeR 3.20.9 package to determine differentially expressed genes (based on fold change >1 and FDR < 0.05) (*Robinson et al., 2010*).

For single-cell RNA sequencing data from cultured fibroblasts, Illumina basecall files (*.bcl) were converted to FASTQ files using Cell Ranger v1.3, using the command-line tool *bcl2fastq* v2.17.1.14. FASTQ files were then aligned to mm10 genome and transcriptome using the Cell Ranger v1.3 pipeline, which generates a gene versus cell expression matrix. The data were analyzed using Seurat v3 (*Stuart et al., 2019*) using the same pipeline and parameters as described above, unless stated below. Given the high average number of features, cells with less than 200 and more than 8500 transcripts were filtered out, obtaining 1121 cells. Data were normalized and scaled as described above. PCA dimensionality reduction was calculated on 50 principal components; UMAP dimensional reduction was calculated on 28 dimensions (value chosen based on the *ElbowPlot* of the standard deviations of the principal components).

qPCR cDNA synthesis of RNAs used for the microarray was performed using the Superscript VILO kit (Invitrogen) following the manufacturer's instructions. PCR reactions were performed using GoTaq Green master mix (Promega). qPCR reactions were performed using SYBR green master mix (Roche) and analyzed using the LightCycler 480 (Roche). At least two individual experiments in triplicate were performed. We tested several primers for endogenous control (*Tbp*, *Gapdh*, *L13*, *Ppi*, *Actab*, and *Hprt*) and chose *Hprt* for further experiments due to its consistent reproducibility within and among samples (*Figure 1—figure supplement 1*). Primers are described in *Supplementary file 1*. All PCR reactions were performed in triplicates and repeated at least twice per sample. Standard error of the mean is represented in all graphs. Prism v7.0 was used for the generation of graphs and statistics.

## Neonatal mouse cardiomyocyte isolation

The protocol from neonatal cardiomyocyte isolation was adapted from *Argentin et al., 1994*. Hearts were collected from litters of 1–3 days old pups, cut open and transferred to trypsin (1 mg/ml in HBSS with phenol red pH 6.4) for overnight digestion at 4°C. The next day, hearts were subjected to 3 × 5 min digestions with Collagenase II (1 mg/10 ml; Worthington) in HBSS with phenol red. The cell suspension was collected in DMEM containing 10% fetal calf serum (FCS) and passed through a 100 µm cell strainer. After 5 min centrifugation at 1000 rpm, cells were plated in 10 cm dishes. Two rounds of 1-hr preplating were done to remove cells highly adherent to plastic such as fibroblasts, before seeding the cell suspension on plates coated with 1:200 fibronectin (Thermo Fisher) in 0.1% gelatin (Thermo Fisher).

## 2D cocultures

Adult fibroblasts isolated from the heart or kidney of Col1a1-GFP mice were cultured to semiconfluence for 3–5 days, after which they were resuspended and cocultured with mouse neonatal cardiomyocytes at a 4:1 ratio. After 24 hr, media was changed to DMEM containing 2% FCS, videos were recorded, and cells were imaged with an Eclipse Ts2 inverted fluorescence microscope (Zeiss) and fixed with 4% PFA for 10 min at 4°C for further staining.

## Cardiac microtissues

Cardiac microtissues were generated as previously described (*Boudou et al., 2012*; *Hinson et al., 2015*), using polydimethylsiloxane (PDMS) 3D microarrays with 24 microwells containing cantilevers. A suspension of 1.3 million cells, 85% hiPSCs derived cardiomyocytes (iCM) and 15% cardiac or kidney fibroblasts, was loaded on each device and cells were seeded in each well by centrifugation. Two millitissue devices with 48 organoids were used per fibroblasts type. The organoids were imaged and fixed 3 days postproduction with 4% PFA for 15 min at RT. Only tissues uniformly anchored to the tips of the cantilevers were included in further analysis.

## Immunostaining

A solution containing 2% bovine serum albumin (BSA), 2% FCS, 0.1% Triton in phosphate buffered saline (PBS) was used for permeabilization, blocking, and dilution. Primary antibodies used in this study are: KRT14 (MA5-11599, Thermo Fisher, mouse monoclonal, 1:100), TBX20 (MAB8124, Novus Biologicals, mouse monoclonal, 1:200), FOXA2 (ab108422, Abcam, rabbit monoclonal, 1:300), HHEX (MAB83771, R&D System, rabbit monoclonal, 1:100), FOXD1 (TA322737, OriGene, rabbit polyclonal, 1:50), PAX8 (NBP2-29903, Novus Biological, mouse monoclonal, 1:100), RSPO1 (AF3474, R&D Novus, goat polyclonal, 1:50), and TNT (RC-C2, DSHB 1:200). Cells were stained overnight at 4°C, washed in PBS and incubated 1 hr with 1:500 secondary antibodies (Alexa Donkey anti Goat Alexa Fluor 568 A11057, Goat anti Mouse Alexa Fluor 568 A11031, Goat anti Rabbit Alexa Fluor 555 A27017; Thermo Fisher). Nuclei were counterstained with 0.1 µg/ml DAPI (D1306, Thermo Fisher).

For the immunostaining of cardiac microtissues, blocking and permeabilization were achieved with 0.1% Triton, 1% BSA in PBS (PBS-T-BSA) for 8 hr. The same solution was used to dilute primary antibodies: TNT (RC-C2, DSHB 1:200) or MF20 (DSHB, mouse monoclonal, 1:100). Staining was performed overnight under gentle agitation at 4°C. After 3 × 5 min washes in PBS-T-BSA, microtissues were stained with secondary antibody and DAPI for 1 hr at room temperature. After staining, the PDMS devices were pulled out of the 35 mm dish used as support and flipped on glass coverslip for confocal imaging.

## Imaging

Immunofluorescence images were acquired using either the either the upright fluorescent microscope Axio Imager.Z2 (Zeiss) or the SP8 confocal microscope equipped with a White Light Laser (Leica). For the cardiac microtissues, 3–4 tiles and 20–34 Z-stacks were imaged per sample. Tiles were combined using the LeicaX confocal software. Z-stack projections and analysis were performed using Fiji version 1.0 and Imaris 8.4.1.

## Cell transfection and crispr knock-down

Cardiac fibroblasts from *ROSA*$^{Cas9-EGFP}$ mice were transfected with guide RNAs using Lipofectamine MessengerMAX (Thermo Fisher) according to the manufacturer's instructions. Briefly, 3 days postisolation, CD45−CD31−CD90 + cells were FACS sorted and replated at about 10,000 cells/cm². After 6 days, when reaching 80–90% confluency, cells were incubated for 5 min with the RNA (1:50)–lipid (1:33) complex in Opti-MEM for 10 min at room temperature. Media was changed after 48 hr and cells were collected for RNA isolation at 72 hr. Two guide RNAs, designed and synthesized in-house (JAX Genetic Engineering Technologies facility), were used for each of the target genes (*Figure 5—source data 1*). CleanCap mCherry mRNA (TriLink Biotechnologies), and guide RNAs for GFP and scrambled guides were used as controls. Guide RNA for the GFP gene were same as in *Li et al., 2014*.

## Cell transplantation in the kidney capsule

Adult fibroblasts from heart, tail, and kidney were isolated from 10-week-old *ROSA* $^{mT/mG}$ male mice as described above. After 10 days cells were collected, counted and 4–5 × 10⁵ cells were transferred to individual 1.5 ml Eppendorf tubes (one per each kidney transplant), resuspended in 15–20 µl of saline solution and kept on ice until surgery. Remaining cells were replated (5 × 10⁴/well; 6-well plate) for cultured cell controls. Syngeneic 10–11 weeks old C57BL6/J mice were used as cell recipients. Mice were anesthetized with 400 mg/kg tribromoethanol intraperitoneally. In parallel, cell suspensions were spotted on a Petri dish and fragments of sterile absorbable gelatin foam (about 1 mm long; Surgifoam, Ethicon) were immersed in the drop.

Fur was removed from the left flank of the animal and eye ointment was applied. The mouse was placed in right lateral recumbency, and a drape positioned over the surgical site. A 6–9 mm skin incision was made parallel and ventral to the spine and midway between the last rib and the iliac crest. A similar incision was made in the underlying abdominal wall. The kidney was externalized by placing forceps under the caudal pole and gently lifting through the incision, kept moist with warm sterile saline. A small incision was made in the capsule over the caudal–lateral aspect of the kidney, and a shallow subcapsular pocket was made with a blunt probe advanced toward the cranial pole of the kidney. The foam previously soaked in the cell suspension was placed in the far end of the subcapsular pocket. If needed, additional foam was used to close the incision site. Absorbable 6.0 Vicryl sutures (Ethicon) were used to close the abdominal wall, 6.0 Vicryl black sutures (Ethicon) for the skin. 0.1% Bupivacaine was applied topically on the injection site and 0.05 mg/kg SR buprenorphine (Zoopharm) was injected subcutaneously.

## Data availability

Sequencing data have been deposited in GEO under accession codes GSE98783 for the microarray experiment, and GSE175765 for the bulk RNAseq datasets generated for the CRISPR experiment, kidney capsule transplant experiment and human cardiac biopsies comparison. Single-cell RNA sequencing data of primary cultured fibroblasts have been deposited in SRA with the identifier SRR5590304. The scRNAseq stromal cell dataset from the Mouse Cell Atlas (Figure 6 in *Han et al., 2018*) was kindly provided by Dr. Guoji Guo's lab. Source Data files have been provided for *Figures 1, 2, 4 and 5*. The code used for the manuscript is currently located at https://github.com/Ramialison-Lab-ARMI/MultiFibroblasts, (*Forte, 2021* copy archived at swh:1:rev:89df5f3cdc503ee4fe603ea956d935b48dc6b669).

## Acknowledgements

We gratefully acknowledge Dr. Guoji Guo's lab for providing the scRNAseq stromal cell dataset, and the contribution of Luis E Lima, Heidi Munger, Philipp Heinrich, and the Genome Technology, Single Cell, Light Microscopy, Flow cytometry, and Computational Sciences Services at The Jackson Laboratory for service provided. This work is supported by grants from the Australian Research Council (ARC) and the National Health and Medical Research Council (NHMRC) to NAR, MWC, MR, and NHMRC/ Heart Foundation Fellowship to MR; and by the JAX Director's Innovation Fund, and the NIH/NIGMS (2 P20 GM104318), the NIH/NIA (5 U01 AG022308-17), and grant from the Leducq Foundation for Cardiovascular Research to NR. The Australian Regenerative Medicine Institute is supported by grants from the State Government of Victoria and the Australian Government. The Jackson Laboratory scientific services are supported by the NIH/NCI (5 P30 CA034196).

## Additional information

### Funding

| Funder | Grant reference number | Author |
|---|---|---|
| Australian Research Council | | Nadia A Rosenthal |
| National Health and Medical Research Council | | Mirana Ramialison<br>Mauro W Costa<br>Nadia A Rosenthal |
| Heart Foundation | | Mirana Ramialison |
| Jackson Laboratory | | Nadia A Rosenthal |
| National Institutes of Health | | Nadia A Rosenthal |
| Leducq Foundation for Cardiovascular Research | | Nadia A Rosenthal |
| Australian Government | | Nadia A Rosenthal |

| Funder | Grant reference number | Author |
|---|---|---|
| State Government of Victoria | | Nadia A Rosenthal |
| Innovation Fund | P20 GM104318 | Nadia A Rosenthal |
| NIH/NIA | U01 AG022308-17 | Nadia A Rosenthal |
| NIH/NCI | P30 CA034196 | Nadia A Rosenthal |

The funders had no role in study design, data collection, and interpretation, or the decision to submit the work for publication.

## Author contributions

Elvira Forte, Conceptualization, Data curation, Formal analysis, Investigation, Methodology, Project administration, Validation, Visualization, Writing – original draft, Writing – review and editing; Mirana Ramialison, Conceptualization, Data curation, Formal analysis, Funding acquisition, Investigation, Methodology, Visualization, Supervision, Visualization, Writing – original draft, Writing – review and editing; Hieu T Nim, Data curation, Formal analysis, Investigation, Methodology, Visualization, Validation, Visualization, Writing – original draft, Writing – review and editing; Madison Mara, Rachel Cohn, Sandra L Daigle, Investigation; Jacky Y Li, Data curation, Formal analysis; Sarah Boyd, Edouard G Stanley, John Travis Hinson, Formal analysis, Supervision; Andrew G Elefanty, Resources, Supervision; Mauro W Costa, Data curation, Formal analysis, Funding acquisition, Investigation, Supervision, Validation, Visualization; Nadia A Rosenthal, Funding acquisition, Supervision, Writing – original draft, Writing – review and editing; Milena B Furtado, Conceptualization, Data curation, Formal analysis, Investigation, Methodology, Project administration, Supervision, Validation, Visualization, Writing – original draft, Writing – review and editing

## Author ORCIDs

Elvira Forte ![ORCID] http://orcid.org/0000-0002-5555-9122
Mirana Ramialison ![ORCID] http://orcid.org/0000-0001-6315-4777
Milena B Furtado ![ORCID] http://orcid.org/0000-0003-1387-325X

## Ethics

Human subjects: Human samples were obtained through the Sydney Heart Bank (SHB) in Australia. Investigators have not collected patient samples or been privy to patient records.

This study was performed in strict accordance with the recommendations in the Guide for the Care and Use of Laboratory Animals of the National Institutes of Health. All of the animals were handled according to approved Institutional Animal Care and Use Committee (IACUC) protocol (#16010) of the Jackson Laboratory. All surgery was performed under tribromoethanol anesthesia, and every effort was made to minimize pain and suffering.

## Decision letter and Author response

Decision letter https://doi.org/10.7554/eLife.71008.sa1
Author response https://doi.org/10.7554/eLife.71008.sa2

---

# Additional files

## Supplementary files

• Transparent reporting form
• Supplementary file 1. Sequence of all the qPCR primers used in the study.

## Data availability

All data have been made available through public databases, as per statement in main manuscript.

The following datasets were generated:

| Author(s) | Year | Dataset title | Dataset URL | Database and Identifier |
|---|---|---|---|---|
| Ramialison M, Nim HT, Furtado MB | 2017 | Transcriptional profile of organ fibroblasts from adult mice | https://www.ncbi.nlm.nih.gov/geo/query/acc.cgi?acc=GSE98783 | NCBI Gene Expression Omnibus, GSE98783 |
| Nim HT | 2021 | RNA-sequencing transcriptional profile of kidney capsule derived organ fibroblasts from adult mice, CRISPR-knockdown organ fibroblasts from adult mice, and human hearts | https://www.ncbi.nlm.nih.gov/geo/query/acc=GSE175765 | NCBI Gene Expression Omnibus, GSE175765 |
| Jackson Laboratory | 2017 | Tissue fibroblasts conserve an organ molecular identity | https://www.ncbi.nlm.nih.gov/sra/SRR5590304 | NCBI Sequence Read Archive, SRR5590304 |
| Han X, Wang R | 2018 | Mapping Mouse Cell Atlas by Microwell seq | https://www.ncbi.nlm.nih.gov/geo/query/acc.cgi?acc=GSE108097 | NCBI Gene Expression Omnibus, GSE108097 |

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
