## [Editor Report]

The authors aim to show that fibroblasts have a heterogenous transcriptome that is retained throughout their lifetime due to their source of embryonic origin. Of great interest is that compelling evidence is provided that these transcriptional signatures have direct translational consequences. This is shown through coculture experiments, where coculture of cardiomyocytes with non-cardiac fibroblasts impairs integration and contractility, while cardiac fibroblasts integrate with cardiomyocyte cultures to create functional beating tissue. This memory is shown to be malleable: three days post implantation in the renal capsule, explanted fibroblasts largely maintained their original transcriptomic signature, while also showing the onset of adaptation to a new microenvironment. In addition, markers are identified which allow the separation of fibroblasts based on their anatomical origin. Considering the lack of tissue-specific markers for fibroblasts, this is a significant advance.

---

## [Decision Letter]

**Decision letter after peer review:**

Thank you for submitting your article "Adult fibroblasts retain organ-specific transcriptomic identity" for consideration by *eLife*. Your article has been reviewed by 3 peer reviewers, and the evaluation has been overseen by a Reviewing Editor and Paul Noble as the Senior Editor. The reviewers have opted to remain anonymous.

Essential revisions:

*Reviewer #1 (Recommendations for the authors):*

The manuscript has strengths in: Figure 7 transplant data were strong; The authors have provided extensive data; Functional confirmation in cardiac lineage was very convincing. The recommendations are:

1. Verification and gene list. A weakness in the initial gene analysis is there are so many genes and pathways mentioned and used (Figure 2a) that it is difficult to determine why the genes in Figure 2 (b-g) were the ones chosen to be validated. The qPCR validation seems to support the hypothesis that these genes have organ specific expressions but their selection (top gene? Specific pathway node? Candidate gene?) from the initial analysis is unclear. It would help to simplify the schematic for Figure 2a and highlight the specific genes that are being validated. This is further compounded by the figure 3 data, which shows mixed results (PAX8 does not really seem to be expressed in the kidney and FOXD1 seems to have an odd pattern of expression; FOXA2 seems to be expressed in some nuclei and not others of the lung) and non-direct comparison between multiple organs.

2. The clusters of the scRNA-seq from both freshly isolated and cultured fibroblasts seem to be due to the batch effects, as it is not very possible that not a single overlapped cell was identified. The listed organ specific genes in heatmaps were hand-picked? as they are identical. Are there any specific genes between fresh isolated and cultured fibroblasts in each organ? A better suggestion should be listing all the shared maker genes and organ specific genes in both freshly isolated and cultured fibroblasts and discussing a little bit the possible related functions.

3. Immunocytochemistry validation should also include the staining on the negative fibroblasts to confirm the "organ specific markers" in Figure 3. More convincing staining experiments should be on the sections of freshly isolated organs with proper and necessary controls.

*Reviewer #2 (Recommendations for the authors):*

The reviewer has some comments on the data presentation, analysis and overall experimental approach that are listed below:

Data presentation:

1) Figure 6i: -LOG(p-value) should be underneath of the X-axis. It does not make sense to show pathways that are not significant, i.e. -LOG(p-value) < 1.3.

2) Figure 5 —figure supplement 1 and Figure supplement 2 are confusing. Figure5 – S1 shows mainly lung, and Figure 5 – S2 shows heart, and it is not clear why both figures also have some data on the kidney.

Analysis:

1) The authors have added single-cell profiling of fibroblasts from the published mouse dataset (Han et al. 2018). However, it would have been more informative to include instead published datasets on human fetal and adult tissues, as these datasets for the heart, kidney and lung are available.

2) Why was microarray and not RNA-seq has chosen to perform gene expression analysis? It is much better to have RNA-seq data instead of as it makes datasets easier to compare with other published datasets.

Experimental approach:

1) The authors used cultured fibroblasts to avoid contamination from parenchyma cells. However, this approach is not ideal, especially because the authors report that cultured cells present an activated/myofibroblast-like phenotype compared to freshly isolated cells. It might also be helpful to describe more in-depth the activated phenotype upon culture compared to freshly isolated cells. Are these get downregulated upon ectopic transplantation?

2) Tissue-specific functionality of fibroblasts in vitro (3D cardiac tissue function) and in vivo (ectopic transplantation) are performed after just 3 days, which might be not sufficient to reveal more differences or similarities. It might be interesting to have these compared to a prolonged period upon co-culture or transplantation, such as three weeks. The reviewer also finds it surprising that the authors observe the change in HOX genes expression as early as 3 days after transplantation, as one would think this should take longer.

*Reviewer #3 (Recommendations for the authors):*

The authors provide a well-written and comprehensive study of the positional "memory" and specialization of fibroblasts. As such, I have only minor comments. Please find below my part-by-part comments to the manuscript.

Abstract

The final sentence of the abstract does not fit with the narrative of the paper, it is too much of a future perspective.

Discussion

Buechler et al. (Nature, 2021) defined in their fibroblast atlas two universal fibroblast populations, from which tissue-specific fibroblasts appear to be derived. The present manuscript does not provide such a cell type: rather than a continuum of states, discrete organ-specific cell states are found. The authors should comment on this discrepancy. Furthermore, the discussion should integrate the present findings with those of Buechler, who touches upon the concept of spatial fibroblast heterogeneity, making the manuscript a direct extension of his findings.

Line 409 seems like an overstatement, as the concept of fibroblasts as a cell type is not at stake here. Rather, the definition of cell state might require an amendment, taking into consideration anatomical location or the local microenvironment to which a cell is exposed.

Figures

The added value of Figure 1 is lost to me. It does not provide much information and would in my view better be suited for the supplements. Regarding Figure 1, lines 108 to 110 can be removed. The statement that is it surprising for a cell type designated mainly as ECM-producing to be rich in transcripts for nuclear and cytosolar proteins is confusing and should therefore be omitted. A cell's primary function does not mean that transcripts related to that function should by default make up the majority of the transcriptome.

Why are averaged raw signals used in Figure 2, rather than normalized expressions?

Figure 5 contains two clusters which cannot be linked back to the organ of origin. Perhaps regressing out the genes of the Hox cluster and the cell cycle can solve this issue?

Figure 6j is missing the cardiomyocyte signature in the Tbx20 KD, while it is the positive control.

Figure 8c does not have a legend. In 8d it would help to move the word 'FDR' next to the legend's color scale.

Results

The first subtitle mentions metabolic components. As these were part of larger gene sets, such as housekeeping, the cytoskeleton, proliferation, this word should be substituted by e.g. homeostatic. No metabolic profiling was performed.

The description of population "KidneyA" in lines 217 to 221 aims to explain a relatively higher expression of genes related to in the response to injury, linking this to tubular cells acquiring mesenchymal phenotypes in vivo. Considering the relative size of the cluster, this seems unlikely to me. It would be of interest to generate a module score for the genes associated to the collagenase-response described by O'Flanagan et al. (Genome Biology, 2019) and test if this cluster is not simply a group of fibroblasts with a stronger reaction to the enzymatic digestion. This can be done easily in Seurat v3 using the AddModuleScore() function.

Line 276: the use of the abbreviation 'HF' for heart failure is not defined. Considering that the abbreviation eHF is coined a few lines earlier to refer to heart fibroblasts, this leads to possible confusion.

Materials and Methods

Lines 493 and 610 mention 10cm dishes, which I believe should be 10 cm² dishes.

Lines 488 and 607 do not mention the basal medium in which the digestion took place.

Line 491 does not state the percentages of sodium pyruvate and pen/strep in the medium.

Line 493 contains a typo in the TrypLE.

Line 541: "log2 transformation and quantile normalization was done using R scripts and public Bioconductor packages". This is too vague to be reproducible, please elaborate and cite the packages used for the analysis.

Line 578: a citation of edgeR is missing (Robinson, Bioinformatics, 2010).

[Editors' note: further revisions were suggested prior to acceptance, as described below.]

Thank you for resubmitting your work entitled "Adult mouse fibroblasts retain organ-specific transcriptomic identity" for further consideration by eLife. Your revised article has been evaluated by Paul Noble (Senior Editor) and a Reviewing Editor.

The manuscript has been improved but there are some remaining issues that need to be addressed, as outlined below:

Please address the batch effect issue as outlined by reviewer #3 as it impacts the significance of the findings.

*Reviewer #1 (Recommendations for the authors):*

The new version of manuscript has satisfactorily answered my concerns.

*Reviewer #2 (Recommendations for the authors):*

The authors have addressed all my comments, and I have no additional comments.

*Reviewer #3 (Recommendations for the authors):*

The authors have adequately addressed most of my comments in both the revisions and their responses. One issue which does persist in my opinion, and which is also mentioned by Reviewer 1 is the grouping of the cells in the scRNA-seq analysis. From a biological perspective at least some overlap would be expected in an analysis only including one cell type. However, there is a very strong separation based on organ, highly resembling a batch effect. In their response to Reviewer 1, the authors mention this could not be possible as all samples were sequenced in the same 10x lane. I believe that by this they mean the same 10x chip, as running 8 samples in the same lane would require additional multiplexing. Regardless, even within the same chip, some technical variation can be expected between lanes.

On this topic the authors also refer to figure 6 of the Mouse Cell Atlas (MCA) paper, but this does not make sense as different dimensionality reduction methods were used between these two papers. The MCA uses tSNE projection, which prioritizes finding local communities, while in this paper UMAP was used, which focuses preserving the global structure of the data. A side effect of the mathematical differences between these methods is that intercluster distance does not have much meaning in tSNE space, but reflects variation in a UMAP projection. Consequently, separation by organ in a UMAP projection as seen here does indeed support the claim of the manuscript. However, the strong separation of samples derived from the same organ (e.g. the two kidney samples) undermines this reasoning, as it implies the presence of a batch effect aiding in the separation. This is not sufficiently addressed by the authors.

– The caption of Figure 2 says b-g, while this should be b,c.

– Figure 7 is labelled a-i, l-m. The panel letters should be updated.

---

## [Author Response]

Essential revisions:Reviewer #1 (Recommendations for the authors):1. Verification and gene list. A weakness in the initial gene analysis is there are so many genes and pathways mentioned and used (Figure 2a) that it is difficult to determine why the genes in Figure 2 (b-g) were the ones chosen to be validated. The qPCR validation seems to support the hypothesis that these genes have organ specific expressions but their selection (top gene? Specific pathway node? Candidate gene?) from the initial analysis is unclear. It would help to simplify the schematic for Figure 2a and highlight the specific genes that are being validated. This is further compounded by the figure 3 data, which shows mixed results (PAX8 does not really seem to be expressed in the kidney and FOXD1 seems to have an odd pattern of expression; FOXA2 seems to be expressed in some nuclei and not others of the lung) and non-direct comparison between multiple organs.

After establishing that the fibroblasts retain a positional identity as shown by the distinct Hox genes expression patterns (Figure 1), we performed differential expression analysis and considered genes that were enriched by 10-fold or more, in single organ fibroblasts compared to tail fibroblasts. Gene Ontology annotation of these organ-enriched genes revealed terms associated with organ development programs; genes in those development related GO terms are shown in the Cytoscape plot (Figure 2 of the re-submission); we have clarified this point in the figure captions (Figure 2, 3) and text. The expression of developmental genes was validated by qPCR on bulk RNA extracted from sorted primary cultured adult fibroblasts and from embryonic fibroblasts (E16.5).

In Figure 3 we validated the expression of the same genes at the protein level using immunocytochemistry. As the reviewer rightly pointed out, the expression of markers like Pax8 and Foxa2 seem to be restricted to certain fibroblasts subtypes (i.e., Pax8 expression is mostly observed in GFP low kidney fibroblasts; Foxa2 nuclear expression seems restricted to smaller lung fibroblasts, not observed in myofibroblast-like cells), showing that despite the selection for CD45- CD31- CD90+ cells, we are still working with populations of fibroblasts in different states. Therefore, we confirmed the expression of the organ-enriched gene signature in scRNAseq data from freshly isolated fibroblasts.

2. The clusters of the scRNA-seq from both freshly isolated and cultured fibroblasts seem to be due to the batch effects, as it is not very possible that not a single overlapped cell was identified. The listed organ specific genes in heatmaps were hand-picked? as they are identical. Are there any specific genes between fresh isolated and cultured fibroblasts in each organ? A better suggestion should be listing all the shared maker genes and organ specific genes in both freshly isolated and cultured fibroblasts and discussing a little bit the possible related functions.

The listed organ-specific genes were predefined based on the signature obtained in the prior network analysis; hence they are identical between the two queried scRNA-seq datasets. The rationale behind the choice is that the core genes preserved under culture conditions are most likely the strongest molecular signature for the cell type. Therefore, we kept looking for this signature in further experiments.

We can exclude a significant batch effect, as the freshly isolated cell data (same dataset subset used in Figure 6 of Mouse Cell Atlas papers) and cultured fibroblasts data (generated by this paper, using fibroblasts cultured under the same conditions and captured in one lane of the 10xChromium) were analyzed separately, and no data integration was performed. Performing data integration would indeed cause batch effects and the resulting bias would outweigh the limited additional insights.

Genes shown in Figure 5b/5e are shared between freshly isolated and cultured fibroblasts. We have provided the relevant function of these organ specific genes (Figure 3—figure supplement 2-7).

3. Immunocytochemistry validation should also include the staining on the negative fibroblasts to confirm the "organ specific markers" in Figure 3. More convincing staining experiments should be on the sections of freshly isolated organs with proper and necessary controls.

We thank the reviewer for this comment. We have not included negative controls in the figure previously, but all controls were performed for every experiment. We have now added the negative staining images (secondary antibody only) for the immunocytochemistry experiments on primary *Col1a1-GFP*+ fibroblasts (Figure 3—figure supplement 1).

The rationale behind performing the staining on primary isolated cells is that parenchymal cell staining can confound visualization of results. For example, Tbx20 is also expressed by other cell types of the heart, including cardiomyocytes and endothelial cells. Performing staining in isolated fibroblasts, clearly displays the overlap between GFP+ fibroblasts and the desired antibody. We are confident the stainings are not artifactual as: 1. primary antibodies show expected expression patterns; 2. negative controls are clean and 3. the protein expression data provided here matches our single cell RNA data analyses.

Reviewer #2 (Recommendations for the authors):The reviewer has some comments on the data presentation, analysis and overall experimental approach that are listed below:Data presentation:1) Figure 6i: -LOG(p-value) should be underneath of the X-axis. It does not make sense to show pathways that are not significant, i.e. -LOG(p-value) < 1.3.

This has been fixed.

2) Figure 5 —figure supplement 1 and Figure supplement 2 are confusing. Figure5 – S1 shows mainly lung, and Figure 5 – S2 shows heart, and it is not clear why both figures also have some data on the kidney.

Figure 5 is now Figure 4.

Supplement 1 and 2 include heatmaps to highlight the differential gene expression among the subclusters of fibroblasts from the same organ. From the freshly isolated stromal cells aggregate (Mouse Cell Atlas 2018), only lung and kidney showed multiple sub-clusters, which are further explored in Supplement 1 (Figure 4 —figure supplement 1). From the in-house scRNAseq dataset on combined cultured fibroblasts, only heart and kidney showed multiple subclusters (Figure 4 —figure supplement 2).

Please note that the analyses are unbiased, so the number of sub-clusters found per organ has not been pre-defined by us. For both studies (fresh isolated cells re-analyzed and cultured cells run in house) we input cells from various organs (kidney, lung, heart, etc).

Analysis:1) The authors have added single-cell profiling of fibroblasts from the published mouse dataset (Han et al. 2018). However, it would have been more informative to include instead published datasets on human fetal and adult tissues, as these datasets for the heart, kidney and lung are available.

The original idea of the manuscript is to demonstrate a core molecular signature kept to adulthood. Embryonic analysis is complicated since organ fibroblasts are only nascent at late fetal time-points.

However, to satisfy the reviewer’s request we have now performed additional analyses of two longitudinal datasets in the mouse:

a. lung E16.5 and E17.5 (https://www.ncbi.nlm.nih.gov/geo/query/acc.cgi?acc=GSE156329, https://doi.org/10.1016/j.isci.2021.102551)

b. kidney E15.5, E17.5, P0 (https://www.ncbi.nlm.nih.gov/geo/query/acc.cgi?acc=GSE149134, https://doi.org/10.1016/j.ydbio.2020.11.002).

We analyzed these datasets separately without data integration, due to the different samples and sequencing conditions. The lung data confirmed the presence of our adult lung fibroblast signature (e.g. Foxf1, Bmp3; left panel of Author response image 1) in the Col1a2+;Vim+ subpopulation of embryonic lung fibroblast (~5500 cells). Thy1 was not significantly expressed at this stage in lung cells, therefore this marker was excluded from the analysis. The time-course expression further revealed the dynamic change of this signature over two developmental time-points. Confirmation was obtained using Module Score analysis (right panel; ‘lung embryonic fibroblasts’ violin plots, generated by the *AddModuleScore*() function in Seurat v3 with default parameters), where the lung fibroblast signature was above the expression of n=100 randomly selected control genes (dashed line) for a subpopulation of lung fibroblasts at E16.5 and E17.5. The module score of E17.5 fibroblasts (3 replicates in different colors – green, cyan and purple) is higher than that of E16.5 fibroblasts (red), suggesting cell maturation plays a role in cementing the adult core molecular code.

**Author response image 1. sa2fig1:** 

Similarly, the embryonic longitudinal kidney data for Thy1+, Col1a2+, Vim+ (~1000 cells) confirmed the adult kidney fibroblast signature was present in embryonic fibroblasts (e.g. Pax8; left panel in Author response image 2). Similar to the embryonic lung fibroblast data, the aggregated expression of the kidney fibroblast signature was above the expression of n=100 randomly selected control genes (dashed line) for a subpopulation of kidney fibroblasts at E15.5 and E17.5 (right panel; ‘kidney fibroblasts’ module score violin plots, generated by the *AddModuleScore*() function in Seurat v3 with default parameters). Once again, the module score of E17.5 is higher than that of E15.5.

2) Why was microarray and not RNA-seq has chosen to perform gene expression analysis? It is much better to have RNA-seq data instead of as it makes datasets easier to compare with other published datasets.

The reason is historical. The microarray data has been performed many years ago and is publicly available but remains unpublished. At the time, microarray technology in mice was mature and high-quality probes were fully established while the RNA-seq technology was emerging. The generated data were robust for cross-tissue interrogation, network analysis, and Hox code analysis. We rationalized that the original bulk microarray data was high quality and we deemed unnecessary to repeat it using RNA-seq posteriorly. However, all data further generated for this manuscript has utilized RNA-seq.

Experimental approach:1) The authors used cultured fibroblasts to avoid contamination from parenchyma cells. However, this approach is not ideal, especially because the authors report that cultured cells present an activated/myofibroblast-like phenotype compared to freshly isolated cells. It might also be helpful to describe more in-depth the activated phenotype upon culture compared to freshly isolated cells. Are these get downregulated upon ectopic transplantation?

Myofibroblasts and their markers are very well characterized in the literature^1,2^. Based on the GEO terms found in this work and previous literature, cells in culture tend to heterogeneously acquire a myofibroblast phenotype due to growth factors in the serum, including TGFB^3^. On the contrary, freshly isolated cells show a homeostatic phenotype, Acta2 negative^4^.

From the single-cell data analyses of both freshly isolated fibroblasts (Figure 4a-c, Figure 4-Source Data 1, Figure Supplement 1 in the revised version) and primary cells in culture (Figure 4d-f, Figure 4-Source Data 2, Figure Supplement 2 in the revised version), we noticed that, when multiple subtypes were present, the signature genes identified from the microarray analysis of the CD45-CD31- Thy1+ fibroblasts tended to be relatively more expressed in the subpopulation with an activated/myofibroblast phenotype. It is interesting that transplanted cells did show increase in GEO terms related to myofibroblast activation, inflammation and fibrosis (Figure 7—figure supplement 1). Our interpretation is that transplanted cells may see the kidney capsule as a foreign/insult environment.

2) Tissue-specific functionality of fibroblasts in vitro (3D cardiac tissue function) and in vivo (ectopic transplantation) are performed after just 3 days, which might be not sufficient to reveal more differences or similarities. It might be interesting to have these compared to a prolonged period upon co-culture or transplantation, such as three weeks. The reviewer also finds it surprising that the authors observe the change in HOX genes expression as early as 3 days after transplantation, as one would think this should take longer.

The reviewer raises a good point. We thought to analyze the transplanted cells after a short time because those were primary fibroblasts transplanted in syngeneic (non-immunocompromised mice), not pluripotent cells as in the classic teratoma assays^5^. In the abovementioned study, luminescence from Fluor-tagged embryonic stem cells disappeared by one week and reappeared 2-7 weeks after transplantation, suggesting the selection of a few resistant clones. In our case, the short transplantation period reduced the chance of sub-selection of transformed cells and the risk of fusion with recipient cells.

We have not tried to keep them for longer periods, and unfortunately, we are not able to perform additional wet-lab experiments at this stage.

Reviewer #3 (Recommendations for the authors):AbstractThe final sentence of the abstract does not fit with the narrative of the paper, it is too much of a future perspective.

The final sentence has been revised accordingly and now reads as:

“In conclusion, our data reveal that adult fibroblasts maintain an embryonic gene expression signature inherited from their organ of origin, thereby increasing our understanding of adult fibroblast heterogeneity. The knowledge of this tissue-specific gene signature may assist in targeting fibrotic diseases in a more precise, organ-specific manner.”

DiscussionBuechler et al. (Nature, 2021) defined in their fibroblast atlas two universal fibroblast populations, from which tissue-specific fibroblasts appear to be derived. The present manuscript does not provide such a cell type: rather than a continuum of states, discrete organ-specific cell states are found. The authors should comment on this discrepancy. Furthermore, the discussion should integrate the present findings with those of Buechler, who touches upon the concept of spatial fibroblast heterogeneity, making the manuscript a direct extension of his findings.

We have modified the discussion to address this issue, lines 457 to 464.

Buecheler et al.^6^ describe a common fibroblast hierarchy across different tissues, with two universal fibroblasts populations, characterized by distinct spatial localization (adventitial versus parenchymal), generating activated or specialized fibroblasts. However, as they mention in the discussion “It is possible that fibroblast subsets may exhibit additional imprinting by their tissue of residence”. We believe our data do not contradict Buecheler et al.’s findings, but the nature our initial analysis does not allow us to explore the full fibroblasts state continuum. Our aim is to identify tissue-specific fibroblast signature genes, as opposed to a generic signature, although such genes were incorporated into the original Figure 1 (now Figure 1—figure supplement 1b-c; Figure 1-Source Data 1).

However, to answer the point raised by the reviewer, we have generated module scores for each of our tissue-enriched signature gene list and enquired whether these expression signatures can be preferentially enriched in any of the two steady-state pan-tissue fibroblast subpopulations described by the Buecheler et al. dataset (violin plots below). We did not observe any common pattern across the different tissues. In general, gene expression was relatively similar among all clusters, with the exception of gonad and heart, that show a tendency toward a relatively higher expression in the parenchymal cluster, and skin with a relatively higher expression in the adventitial cluster.

**Author response image 3. sa2fig3:** 

Line 409 seems like an overstatement, as the concept of fibroblasts as a cell type is not at stake here. Rather, the definition of cell state might require an amendment, taking into consideration anatomical location or the local microenvironment to which a cell is exposed.

We have toned down this statement as suggested by the reviewer to “suggesting a strong influence of the anatomical location or the local microenvironment on the cell state.” (now lines 456-457).

FiguresThe added value of Figure 1 is lost to me. It does not provide much information and would in my view better be suited for the supplements. Regarding Figure 1, lines 108 to 110 can be removed. The statement that is it surprising for a cell type designated mainly as ECM-producing to be rich in transcripts for nuclear and cytosolar proteins is confusing and should therefore be omitted. A cell's primary function does not mean that transcripts related to that function should by default make up the majority of the transcriptome.

We have edited the figures and manuscript accordingly: Figure 1 has been moved to Figure 1—figure supplement 1b-c. We have removed the last part of paragraph 1 referring to the cellular localization of commonly expressed genes.

Why are averaged raw signals used in Figure 2, rather than normalized expressions?

Figure 2 is now Figure 1. The idea behind using raw signal instead of normalization is that tail can also be a queried sample in this particular figure. This way we could present all tissues in separate plots. Comparisons were only made between the expression of Hox gene members within the same organ, which could be made based on absolute expression levels. In this case, normalization was not required and would not change the ranking between Hox genes within the same tissue.

Figure 5 contains two clusters which cannot be linked back to the organ of origin. Perhaps regressing out the genes of the Hox cluster and the cell cycle can solve this issue?

We believe those clusters are likely due to non-fibroblasts populations, because the single-cell analysis was done on primary cells (cultured for 5-days) sorting only based on the viability markers not on CD45-CD31-Thy1+, as in the initial bulk microarray analyses.

We have already regressed out the cell cycle genes in the presented analysis, but we thought regressing out the Hox genes (very strong in those non-fibroblasts populations) would alter our clustering.

The Hox code signature is very strong in these clusters; regressing Hox genes would dilute all signatures in all clusters. Although we can’t explain what these cells are, we kept them in the plot for transparency.

Figure 6j is missing the cardiomyocyte signature in the Tbx20 KD, while it is the positive control.

The ‘cardiomyocyte signature genes’ we not the positive control. These genes are found in cardiac fibroblasts, albeit at much lower levels than what would be expected for a cardiomyocyte (we estimate ~100 fold or so lower). We have routinely encountered sarcomeric cardiomyocyte genes in cardiac fibroblasts, except for the bona-fide marker myosin heavy chain. We believe that the core transcriptional signature in cardiac fibroblasts generates some spurious low level expression of sarcomeric genes in these cells, although it seems this expression has no biological relevance, considering that cardiac fibroblasts have no sarcomere structures. This reinforces the need to use isolated cultured cells to determine the core transcriptional identity, as carry over RNA could certainly impact such findings.

For the plots in 6j we selected representative genes for each category that were significantly regulated in the KD. Cardiomyocyte signature genes (*Tnnt2, Tnnc1*) were differentially regulated in the Gata4KD but were not differentially expressed in the Tbx20KD.

Figure 8c does not have a legend. In 8d it would help to move the word 'FDR' next to the legend's color scale.

(c) was not in brackets hence it appeared as if 8c did not have a legend. We have corrected this and the corresponding legend is: “Multidimensional scaling plot calculated on the top 500 genes post-normalization, to visualize the level of transcriptomic similarity among all the samples.” Additionally, we have moved FDR near the legend scale.

ResultsThe first subtitle mentions metabolic components. As these were part of larger gene sets, such as housekeeping, the cytoskeleton, proliferation, this word should be substituted by e.g. homeostatic. No metabolic profiling was performed.

The first subtitle has been changed to “Cell homeostasis and extracellular matrix components comprise a generic fibroblast gene signature”.

The description of population "KidneyA" in lines 217 to 221 aims to explain a relatively higher expression of genes related to in the response to injury, linking this to tubular cells acquiring mesenchymal phenotypes in vivo. Considering the relative size of the cluster, this seems unlikely to me. It would be of interest to generate a module score for the genes associated to the collagenase-response described by O'Flanagan et al. (Genome Biology, 2019) and test if this cluster is not simply a group of fibroblasts with a stronger reaction to the enzymatic digestion. This can be done easily in Seurat v3 using the AddModuleScore() function.

We have performed the ‘Module Score’ analysis as per the reviewer’s suggestion. We used the kidney marker genes, as well as the collagenase-response genes (“core 512 genes”), as described in O'Flanagan et al.^7^, to assess the module score based on the Mouse Cell Atlas single-cell data (see panels A, B, C in violin plots in Author response image 4). The data showed that the “KidneyA” population (Panel A) was distinct from the “Kidney B” population (Panel B), based on each corresponding subpopulation marker identified in this study. The same kidney cells however did not show any difference in module scores when using the collagenase-response genes (Panel C), showing that the Kidney A and B sub-populations are not the result of differential reaction to enzymatic digestion. Thus, we maintain our hypothesis that the “KidneyA” cluster corresponds to tubular cells acquiring mesenchymal phenotypes.

**Author response image 4. sa2fig4:** 

Line 276: the use of the abbreviation 'HF' for heart failure is not defined. Considering that the abbreviation eHF is coined a few lines earlier to refer to heart fibroblasts, this leads to possible confusion.

To avoid confusion, HF was replaced with heart failure on line 276.

Materials and MethodsLines 493 and 610 mention 10cm dishes, which I believe should be 10 cm² dishes.

Thanks for pointing this out. This has been corrected.

Lines 488 and 607 do not mention the basal medium in which the digestion took place.

We modified the text to explain that the digestion occurred in Trypsin (line 488) or in Collagenase resuspended in HBSS with phenol red (line 607).

Line 491 does not state the percentages of sodium pyruvate and pen/strep in the medium.

We have added the percentages of sodium pyruvate and pen/strep; we have also added GlutaMAX, which was missing under media composition.

Line 493 contains a typo in the TrypLE.

This has been corrected.

Line 541: "log2 transformation and quantile normalization was done using R scripts and public Bioconductor packages". This is too vague to be reproducible, please elaborate and cite the packages used for the analysis.

The text has been modified to incorporate more methodology details. We also provide all coding used for the manuscript under https://github.com/Ramialison-Lab-ARMI/MultiFibroblasts.

Line 578: a citation of edgeR is missing (Robinson, Bioinformatics, 2010).

This reference has been added to the modified manuscript.

References:

1. Baum, J.B.S., Duffy, H.S. Fibroblasts and Myofibroblasts: What Are We Talking About?, Journal of Cardiovascular Pharmacology (2011). DOI: 10.1097/FJC.0b013e3182116e39.

2. Tarbit, E., Singh, I., Peart, J.N. and Rose'Meyer, R.B. Biomarkers for the identification of cardiac fibroblast and myofibroblast cells. Heart Fail Rev (2019). DOI: 10.1007/s10741-018-9720-1.

3. Siani, A., Khaw, R., Manley, O. et al. Fibronectin localization and fibrillization are affected by the presence of serum in culture media. Sci Rep (2015). DOI: https://doi.org/10.1038/srep09278.

4.Forte, E., Skelly, D.A., Chen, M., et al. Dynamic interstitial cell response during myocardial infarction predicts resilience to rupture in genetically diverse mice. Cell Reports (2020). DOI: https://doi.org/10.1016/j.celrep.2020.02.008

5. Zhu, F.; Sun, B., Wen, Y., et al. Modified Method for Implantation of Pluripotent Stem Cells Under the Rodent Kidney Capsule. (2014). DOI: https://doi.org/10.1089/scd.2014.0099

6. Buechler, M.B., Pradhan, R.N., Krishnamurty, A.T., et al. Cross-tissue organization of the fibroblast lineage (2020). DOI: https://doi.org/10.1038/s41586-021-03549-5

7. O’Flanagan, C.H., Campbell, K.R., Zhang, A.W. et al. Dissociation of solid tumor tissues with cold active protease for single-cell RNA-seq minimizes conserved collagenase-associated stress responses. Genome Biol (2019). DOI: https://doi.org/10.1186/s13059-019-1830-0.

[Editors' note: further revisions were suggested prior to acceptance, as described below.]

Reviewer #3 (Recommendations for the authors):The authors have adequately addressed most of my comments in both the revisions and their responses. One issue which does persist in my opinion, and which is also mentioned by Reviewer 1 is the grouping of the cells in the scRNA-seq analysis. From a biological perspective at least some overlap would be expected in an analysis only including one cell type. However, there is a very strong separation based on organ, highly resembling a batch effect. In their response to Reviewer 1, the authors mention this could not be possible as all samples were sequenced in the same 10x lane. I believe that by this they mean the same 10x chip, as running 8 samples in the same lane would require additional multiplexing. Regardless, even within the same chip, some technical variation can be expected between lanes.

We thank the reviewer for their comment and the opportunity to clarify this further. The primary cultured fibroblasts from different organs were mixed in one single lane of the 10x Chromium chip, without multiplexing. We identified the clusters based on the expression of the organ-specific core genes identified from the initial bulk microarray experiment and validated via ICC, and qPCR.

On this topic the authors also refer to figure 6 of the Mouse Cell Atlas (MCA) paper, but this does not make sense as different dimensionality reduction methods were used between these two papers. The MCA uses tSNE projection, which prioritizes finding local communities, while in this paper UMAP was used, which focuses preserving the global structure of the data. A side effect of the mathematical differences between these methods is that intercluster distance does not have much meaning in tSNE space, but reflects variation in a UMAP projection. Consequently, separation by organ in a UMAP projection as seen here does indeed support the claim of the manuscript. However, the strong separation of samples derived from the same organ (e.g. the two kidney samples) undermines this reasoning, as it implies the presence of a batch effect aiding in the separation. This is not sufficiently addressed by the authors.

We have confirmed the separation between organ fibroblasts observed in our single-cell data by cross-validating with the Mouse Cell Atlas data. Three independent clustering methods were employed (Author response image 5), including UMAP (Panel A), t-SNE (Panel B) and PHATE (PMID:31796933, Panel C), all showing a clear separation between organ fibroblasts. The separation between two kidney subpopulations were also confirmed, which is consistent with renal fibroblasts literature (https://journals.biologists.com/dev/article/147/15/dev190108/143923/Identification-and-characterization-of-cellular; https://www.nature.com/articles/s41467-022-28226-7).

**Author response image 5. sa2fig5:** Visualisation of single-cell organ fibroblasts data from the Mouse Cell Atlas (PMID:29474909), using different clustering methods: (A) UMAP, (B) t-SNE, and (C) PHATE.